# Silver lining to a climate crisis in multiple prospects for alleviating crop waterlogging under future climates

Ke Liu [1,2,29], Matthew Tom Harrison [1,29] ✉, Haoliang Yan [3,29], De Li Liu [4,5], Holger Meinke [1], Gerrit Hoogenboom [6], Bin Wang [4], Bin Peng [7,8,9], Kaiyu Guan [7,8,9], Jonas Jaegermeyr [10,11,12], Enli Wang [13], Feng Zhang [14], Xiaogang Yin [15], Sotirios Archontoulis [16], Lixiao Nie [17], Ana Badea [18], Jianguo Man [19], Daniel Wallach [20], Jin Zhao [21], Ana Borrego Benjumea [18], Shah Fahad [22], Xiaohai Tian [2], Weilu Wang [23], Fulu Tao [24,25], Zhao Zhang [26], Reimund Rötter [27], Youlu Yuan [3], Min Zhu [18], Panhong Dai [28], Jiangwen Nie [15], Yadong Yang [15], Yunbo Zhang [2] & Meixue Zhou [1]

Extreme weather events threaten food security, yet global assessments of impacts caused by crop waterlogging are rare. Here we first develop a paradigm that distils common stress patterns across environments, genotypes and climate horizons. Second, we embed improved process-based understanding into a farming systems model to discern changes in global crop waterlogging under future climates. Third, we develop avenues for adapting cropping systems to waterlogging contextualised by environment. We find that yield penalties caused by waterlogging increase from 3–11% historically to 10–20% by 2080, with penalties reflecting a trade-off between the duration of waterlogging and the timing of waterlogging relative to crop stage. We document greater potential for waterlogging-tolerant genotypes in environments with longer temperate growing seasons (e.g., UK, France, Russia, China), compared with environments with higher annualised ratios of evapotranspiration to precipitation (e.g., Australia). Under future climates, altering sowing time and adoption of waterlogging-tolerant genotypes reduces yield penalties by 18%, while earlier sowing of winter genotypes alleviates waterlogging by 8%. We highlight the serendipitous outcome wherein waterlogging stress patterns under present conditions are likely to be similar to those in the future, suggesting that adaptations for future climates could be designed using stress patterns realised today.

Increasingly frequent and compound extreme weather events driven by the intensification of the global water cycle threaten the sustainability and consistency of agri-food production[1–3]. Coupled with global population growth and a burgeoning demand for food, exposure to weather extremes demand the development of new knowledge, technologies and practices that enable scalable, sustainable intensification[4,5].

Robust projections of climate impacts on crop growth underpinned by process-based models[6,7] are fundamental in the quest to design effective and credible systems-based adaptations that minimise

downside risk associated with future climate[8–10]. Application of such models enables consideration of nonlinear, integrated crop responses to environmental, genetic and management conditions[7,11], supporting the development of socially-acceptable and profitable climate change adaptation and/or greenhouse gas emissions mitigation strategies[12–14]. However, while the overwhelming majority of previous climate change assessments have used a lens focused on either drought, heat or gradual climate change[1,3,15–17], our knowledge of the impacts of soil waterlogging on crop growth is very much in its infancy[18–21].

Globally, around 27% of cultivated lands are impacted by flooding each year, with annual costs of flood damage over the last half-century reaching a headline value of US$19 billion[22–25]. Intensification of the global water cycle called forth by the climate crisis would appear to be driving a higher prevalence of waterlogging, placing pressure on the use-efficiency of economic, natural and social capital[20]. While genotype (G) × environment (E) × management (M) studies pertaining to climate change adaptation abound[26–31], such work is often not conducted in a way that facilitates scaling to other regions or transferability from other studies. Here, we develop a new approach for assimilating manifold results from crop models into common, discrete sets of groups. These groups—characterised by daily stress trajectories plotted over the crop lifecycle as a function of phenology—invoke plant stress, because perceived stress represents an integrated measure of biomass, canopy leaf area, cumulative water supply, vapour pressure deficit and several other factors interacting across an atmosphere-plant-soil continuum. As such, plant stress has long been a ubiquitous target for quantification and manipulation in molecular, breeding and agronomic studies[32–35].

While G × E × M factorial studies are useful, attempts to interpret results using the association between management interventions and maturity biomass or yield[36] can make it difficult to derive functional, rationally bounded[37] insights across all of the interventions deployed. In contrast, we suggest that crop stress patterns characterised as a function of phenology are limited in type; when grouped across an entire factorial analysis, such relationships can be aggregated into common groups and recurrence intervals, even though individual stress trajectories may appear unique. To standardise contrasts across treatments, we grouped waterlogging stress as a function of phenology. We focus on waterlogging stress and barley as case studies, but the principles could be generically applied to any crop or biological variable. A fundamental contribution of our approach is the ability to functionally categorise big datasets. Armed with knowledge of stress prevalence and pattern using this method, scientific practitioners can (1) more intuitively identify the most appropriate adaptation within stress patterns that are more probable for their environment and (2) transfer adaptations across regions within any given stress type[35,38].

Building on foundational insights from our previous waterlogging experiments conducted using a range of genotypes and treatments in controlled environments[39], we enumerate the effects of waterlogging on photosynthesis and phenology and then use these insights to improve the capacity of the internationally renowned model APSIM to simulate the impacts of waterlogging on crop growth[40]. Although past work has shown that our new waterlogging algorithms reproduce the effects of waterlogging stress on contemporary barley genotypes[40] with reasonable precision, the validity of our new algorithms across a broad array of global cropping environments remains unknown. To fill these knowledge gaps, we first calibrated and evaluated the waterlogging-enabled version of APSIM using measured field data from five countries. We then applied the waterlogging-enabled model and novel clustering paradigm in each of the major barley production zones across the world with the specific objectives of (1) quantifying the effects of climate change on waterlogging, (2) characterising common waterlogging stress patterns and frequencies across environments, (3) determining the extent with which common stress patterns change under future climate, and (4), quantifying the extent with which waterlogging tolerance genotypes, genotypic phenology and sowing time mitigate effects of waterlogging under future climates.

## Results and discussion

### Conceptualising impacts of waterlogging on phenology and photosynthesis

Past work has shown that crop sensitivity to waterlogging stress is critically dependent on the developmental stage in which waterlogging occurs[31]. As such, we modelled waterlogging stress as a function of phenology, which is in itself a significant advance on the majority of previous studies, the latter assuming that waterlogging stress is primarily a function of water-filled pore space and has negligible effect on crop ontogeny (e.g., ref. [40]). We developed new functions to account for experimentally observed effects of waterlogging on photosynthesis and phenology (*oxdef photo* and *oxdef pheno*, respectively; Fig. 1a)[40]. Each dimensionless function assumes multipliers ranging from unity to nil in the form of $y = f(x)$, where $y$ is the stress factor and $x$ is soil moisture. When $x$ is at or below field capacity, $y = 1$; $y$ linearly decreases with increasing $x$ until the point at which the soil is saturated ($y = 0$). These functions were incorporated into the APSIM software platform to enable improved simulation of crop responses to waterlogging as part of an integrated system. We calibrated the waterlogging-enabled framework using published data from field observations across five countries (Australia, Argentina, China, Canada and Ireland; Supplementary Table 1). Including the new waterlogging functions significantly improved the performance of APSIM in simulating the biophysical impacts of waterlogging relative to the default version of the model, with the root mean square error (RMSE) for waterlogged yield loss predictions decreasing from 0.3 to 0.1 (Fig. 1b). The modified model adequately captured the variation in grain yield of multiple genotypes in response to a range of waterlogging treatments across environments (Fig. 1b), with simulations accounting for 70% of the variation in observed yield.

### Impacts of a changing climate on global soil waterlogging and barley yield

Using downscaled projections from Assessment Report 6 (AR6[36]) from 27 global circulation models (GCMs; Supplementary Table 2), we quantified how current waterlogging frequencies may change under future climates. Following recent reports[41], we simulated crop growth and development using the most plausible greenhouse gas emissions scenario (ie. SSP585) for climate horizons of 2030–2059 and 2070–2099 (hereafter respectively referred to as 2040 and 2080). To account for variable growing season durations under future climates, we examine crops sown relatively early and late at each site in factorial combination with shorter-growing season genotypes ('spring') and longer growing season genotypes ('winter'; Supplementary Table 3).

Our simulations suggest that even though the risk of severe waterlogging will increase under future climate (2-10% increase across GCMs, sites and sowing dates; Supplementary Fig. 1), yields will also slightly increase due to fertilisation from atmospheric $CO_2$ enrichment and mitigation of cold stress at high latitudes (Fig. 2a–d and Supplementary Fig. 1). Our work suggests that past estimates of yield that do not account for soil waterlogging may be overestimated: here we show that simulated future yields decreased by 8–18% in 2040 and 17–26% in 2080 when physiological effects were embedded in the modelling framework (Fig. 2a, d). This modulating effect of waterlogging on yield was especially pronounced in winter genotypes regions (Fig. 2c), likely because such crops have longer growing seasons and greater annual rainfall. Globally, the average yield penalty caused by waterlogging was 11% for the historical baseline, 14% in 2040 and 20% in 2080 for winter barley (median yield penalty 130–591 kg ha⁻¹; Fig. 4d), while for spring barley yield penalties were 3% for the historical baseline, 6% in 2040 and 10% in 2080 (median yield penalty 50–91 kg ha⁻¹; Fig. 4a) across GCMs, sites and sowing dates.

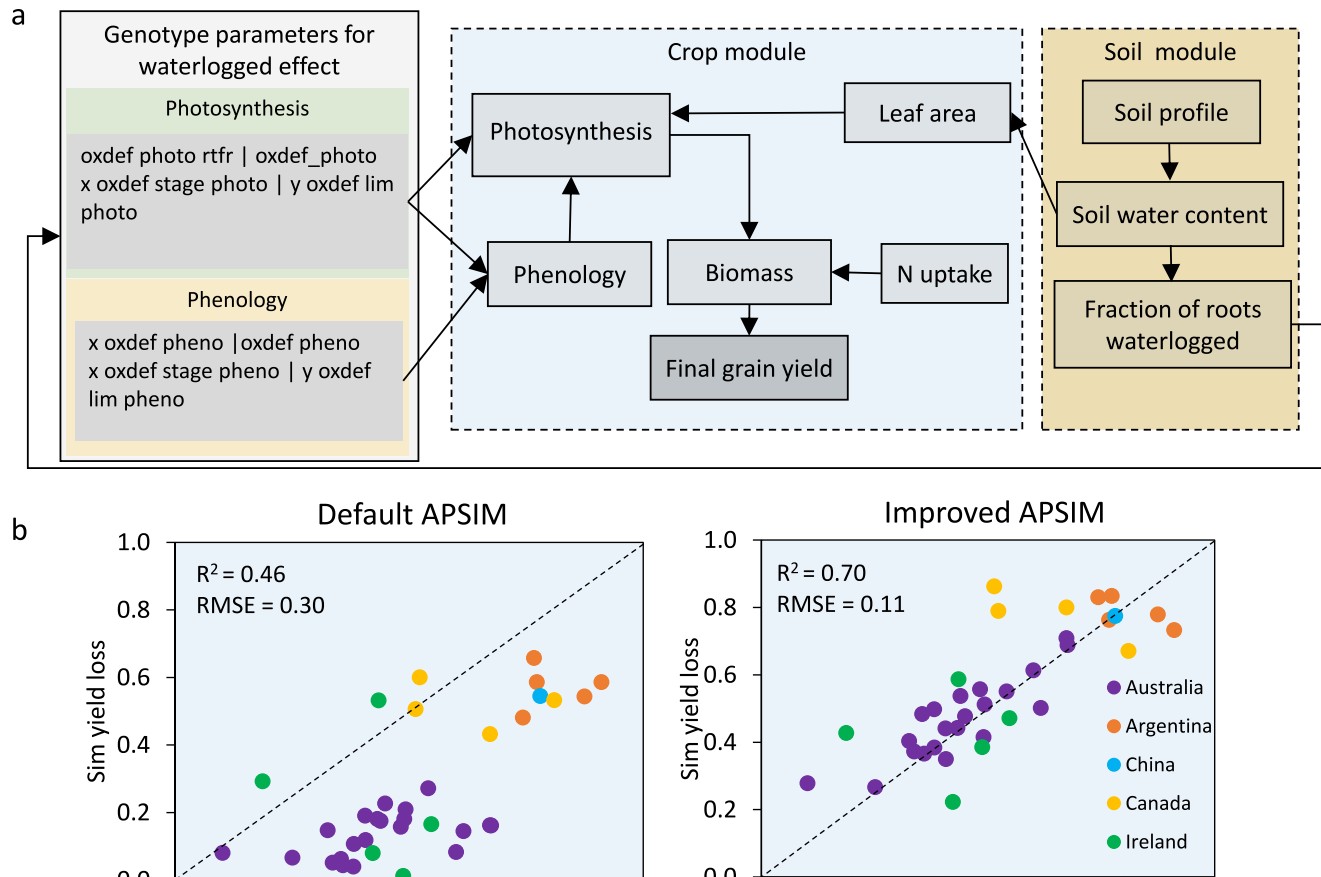

**Fig. 1 | Framework invoked for modelling waterlogging (WL) stress, including conceptual design of crop physiological responses to waterlogging and model evaluation before (default) and after (improved) modification. a** Schematic of genotypic traits influenced by waterlogging and linkage with existing soil and water sub-models in APSIM. **b** Comparison of observed (Obs) and simulated (Sim) waterlogged yield loss compared with controls across environments simulated by improved and default versions of APSIM. Data in (**b**) represent contemporary barley genotypes with varying waterlogging tolerance ($n = 36$). Parameter descriptions are provided in Supplementary Table 4.

Despite increased impacts of waterlogging, spring barley yields increased by 5% in 2040 and 13% in 2080 for early sowing (ES) and by 7% in 2040 and 18% in 2080 for late sowing (Supplementary Fig. 1). Future climates had variable effects on spring barley yield, ranging from positive (e.g., Australia, Germany, Spain, France, United Kingdom, Ukraine and Russia) to antagonistic (Argentina, Canada, Central Ethiopia, Ukraine and United states; Supplementary Table 4). Averaged across sites and climate horizons, yields increased by 22% and 9% for early and late sown winter barley (Supplementary Fig. 1). For both future climate horizons, winter barley yields increased for most regions under early sowing, with greater gains expected in Europe (18%; Supplementary Table 4). These changes suggest that forward shifts in sowing time of long-season genotypes may benefit yields, congruent with other work[42].

**Distilling common stress patterns across diverse environments, genotypes and management approaches**

Improved understanding of common waterlogging-stress seasonal patterns allows insight into the timing of waterlogging stress relative to crop phenology, which then governs cumulative effects on growth, tillering, floral development and yield[35,40,43]. When applied in the present study, these results help explain differences between yield penalties caused by waterlogging stress between winter and spring barley (Fig. 3a–f). Using waterlogging stress outputs from the model computed as a function of historical climate, soil physics, atmospheric

demand, plant biology and agronomy, we calculate stress indices for each day of crop growth.

We applied unsupervised *k*-means clustering to many thousand individual trajectories of discretised waterlogging stress as a function of the phenological stage into four common clusters (Figs. 3 and 4); within each stage, the algorithm minimises within-cluster variances. The four clusters accounted for 71% of the variance for spring barley and 80% for winter barley (increasing to five clusters accounted for 74% and 85% of total variance for spring and winter barley and was deemed superfluous accuracy; Supplementary Fig. 2). While we showcase barley and waterlogging stress as exemplars, the principles shown here could be applied to any crop, region, stress type or biophysical model output.

Winter genotypes experienced substantially different patterns of seasonal waterlogging stress relative to spring genotypes at the global scale (cf. Fig. 3a–f); waterlogging primarily occurred in the juvenile phase of winter barley (WW3) cf. during reproductive development of spring types (SW2-3). While cereals are more likely to experience yield losses when exposed to waterlogging during their reproductive phases (yield formation of cereals being tightly coupled with kernel number and mass) we showed that winter genotypes exposed to waterlogging during their juvenile phases had lower yields than spring genotypes exposed to waterlogging during their reproductive phases, because the magnitude of waterlogging experienced by winter types was greater. Put another way, yield penalties caused by waterlogging

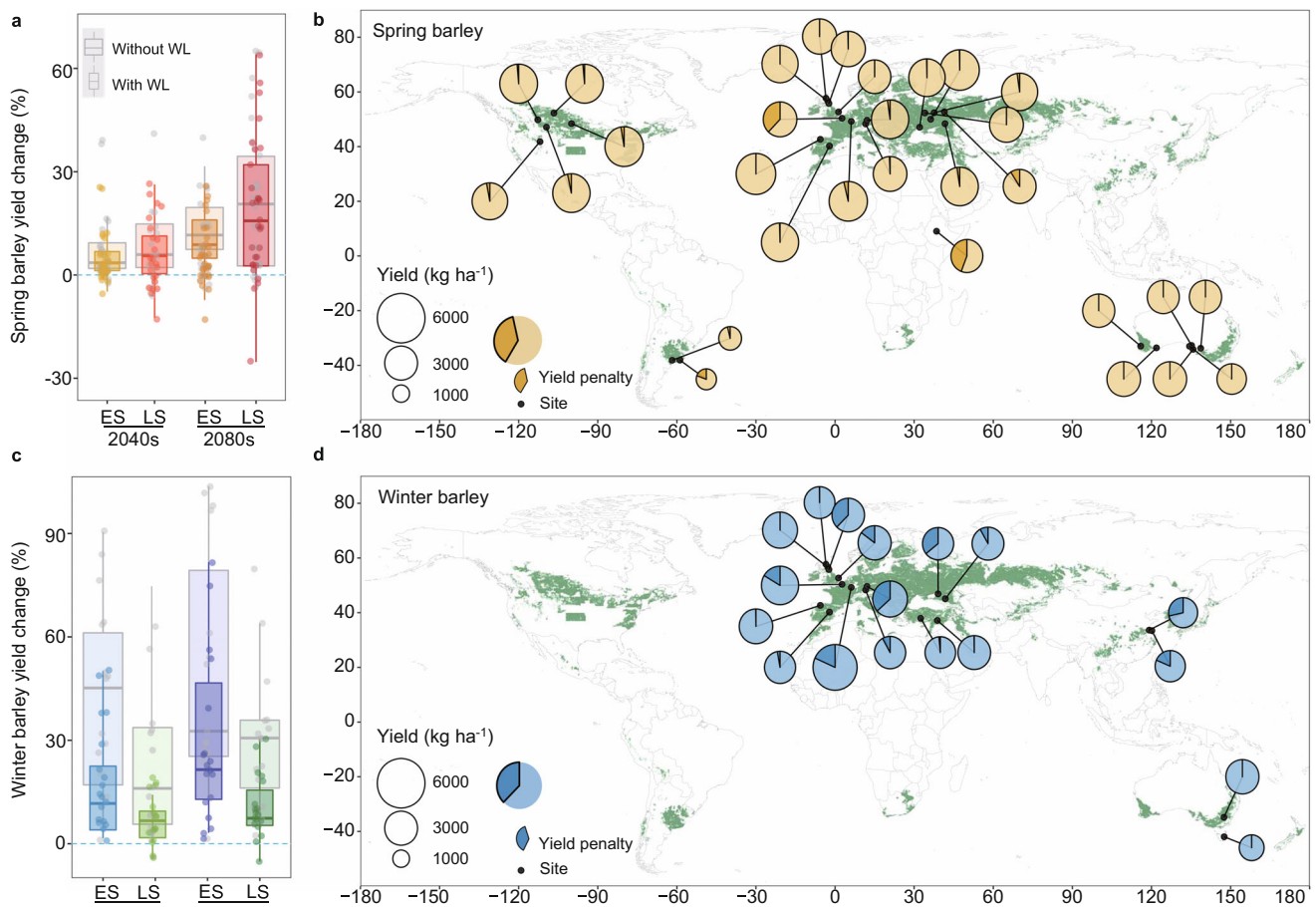

**Fig. 2 | Impacts of waterlogging on yield under future climates (2040, 2080) relative to the historical baseline (1985–2016) for early and late sowing (ES, LS).** **a**, **c** Simulated yield differences under future climates with and without waterlogging (WL) for genotypes with early (spring sowing barley) or late maturity (autumn/winter sowing barley). **b**, **d** Simulated yields (pie charts; dark segments denote yield penalty) under late sowing for spring barley and early sowing for winter barley in 2040 (results for early or late sowing in 2040 and 2080 can be found in supplementary Fig. 12). Yields were simulated with APSIM using downscaled projections from 27 GCMs ($n = 27$). Boxplots indicate simulated yield change across sites and GCMs; box boundaries indicate 25th and 75th percentiles, whiskers below and above each box denote the 10th and 90th percentiles, respectively. Green regions in the maps define predominant barley cropping areas. The map was modified using *R* package ggplot2'maps (version 3.4.0)' with the Natural Earth dataset in a publica domain (https://www.naturalearthdata.com).

reflected an important trade-off between the duration of waterlogging experienced within a given phase and the timing of waterlogging relative to the crop stage; across simulations, yield penalties associated with winter barley were more severe than those of spring barley (Fig. 4a, d).

While recurrence frequencies for each of the four main waterlogging stress patterns for spring genotypes remained similar under future climate, frequencies of early severe (WW3) and mild (WW1) waterlogging during the juvenile phases of winter genotypes increased under future climate at the expense of seasons with minimal waterlogging. Stress pattern WW1 increased from 7% to 17% (under early sowing; Fig. 4e) while WW3 from 3% to 8% (under late sowing; Fig. 4f) compared with the baseline and 2080 periods (Fig. 3). Increased frequencies of severe waterlogging underpin the greater reductions in yields observed for winter genotypes compared with spring genotypes under future climate (Figs. 2a, c and 4d), primarily due to increased waterlogging in France, the UK, Russia and China (Supplementary Fig. 3).

**Pathways for adapting agricultural systems to waterlogging**
Adaptation of agricultural systems to climate change has and will require cross-disciplinary action: new knowledge, practices and technologies that integrate agronomic, environmental, molecular, social and institutional dimensions will be required[5,44,45]. By 2080, early

sowing of spring barley reduced the occurrence of low waterlogging (SW0; Fig. 4b), while later sowing of spring barley increased the likelihood of low waterlogging occurrence but did not affect the frequency of the most severe type of waterlogging SW3 (Fig. 4c). In contrast, earlier sowing of winter barley diminished frequencies of both severe and low waterlogging stress (WW1 and WW3; Fig. 4e), while later sowing of winter types increased risk of early-onset severe and moderate waterlogging (WW1-WW3; Fig. 4f). Overall, we suggest that sowing time of spring barley in 2080 had relatively little effect on the magnitude of the type of waterlogging stress, while later sowing of winter barley was likely to increase the likelihood of exposure to waterlogging stress.

Altering sowing time coupled with the adoption of superior genetics resulted in further gains in yield. Based on experimental observations, we developed in silico genotypes tolerant to soil hypoxia and anoxia typically experienced when soils become waterlogged[46]. After verifying the ability of the improved model to capture behaviour of tolerant genotypes during and after waterlogging (Fig. 1b), we examined the long-term performance and yield benefit expected when waterlogging tolerant spring and winter genotypes were coupled with other prospective adaptations (altered sowing time and/or phenological duration). New genotypes with waterlogging tolerance demonstrably increased barley yield under wetter years (Fig. 5) and in general (Supplementary Fig. 4) under future climates. Across sites, the average

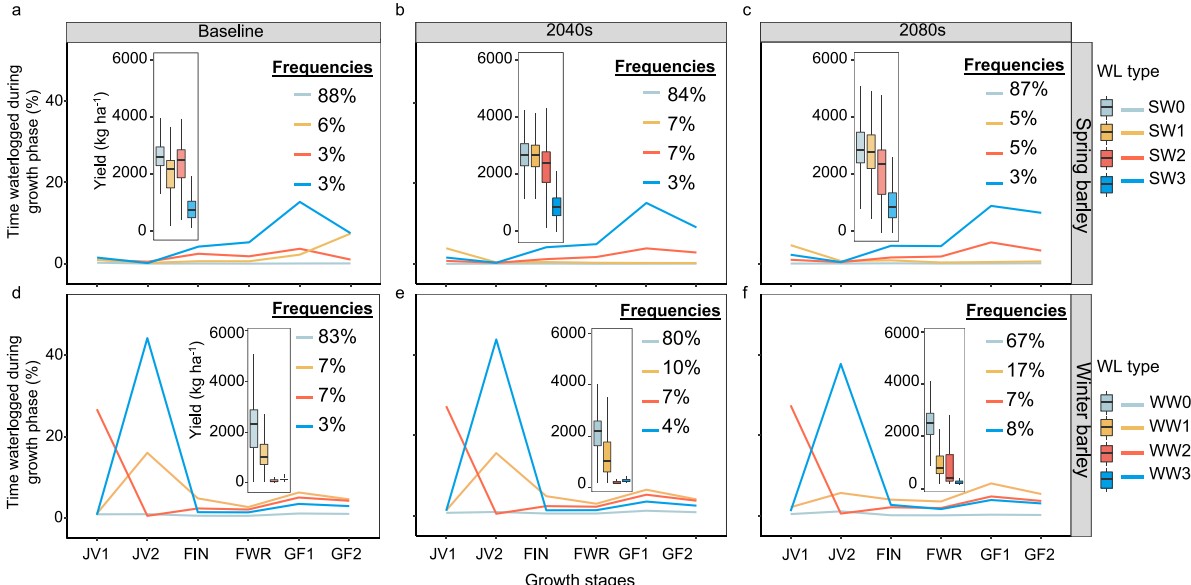

**Fig. 3 | Waterlogging (WL) stress patterns and frequencies and grain yields for the baseline (1985–2016), 2040 (2030–2059) and 2080 (2070–2099).** Data shown for spring (**a**–**c**) and winter barley (**d**–**f**) across sites, sowing times and genotypes. Four key waterlogging stress patterns across sites and genotypes are depicted: stress patterns for spring barley include SW0 (minimal waterlogging); SW1 (low moderate-late waterlogging); SW2 (late-onset moderate waterlogging); SW3 (late-onset severe waterlogging) and winter barley WW0 (minimal waterlogging); WW1 (low early-onset waterlogging relieved later); WW2 (moderate early-onset waterlogging); WW3 (severe early-onset waterlogging). Boxplots indicate grain yields for spring and winter barley across sites and GCMs; box boundaries indicate the 25th and 75th percentiles across 27 GCMs, whiskers below and above the box indicate the 10th and 90th percentiles. Growth stages include the early juvenile phase (JV1, 10 <= APSIM growth stage <21); late juvenile phase (JV2, 21 <= APSIM growth stage <32); floral initiation to heading (FIN, 32 <= APSIM growth stage <65); flowering to grain filling (FIN, 65 <= APSIM growth stage <71; early grain filling (GF1, 71 <= APSIM growth stage <80) and late grain filling (GF2, 80 <= APSIM growth stage <87).

yield benefit of waterlogging tolerant lines was 14% and 18% (s.d., 23% and 34%) for early- and later sowing in the 2040 s compared with the baseline genotypes. Similar yield benefits were observed in 2080 (Supplementary Fig. 5). Mean yield benefits were greater for winter genotypes (480–620 kg ha$^{-1}$) than spring genotypes (194–213 kg ha$^{-1}$; Fig. 5). Importantly, yield benefits associated with waterlogging tolerance of new genotypes did not come at the expense of yield in drier years, and reduced downside risk associated with low yielding years (Supplementary Fig. 4).

Our results suggest that there would be more scope for and potential impact of waterlogging tolerant genotypes in environments with longer, cooler and more temperate growing seasons (e.g., the UK, France, Russia and China; Fig. 5 and Supplementary Fig. 6), compared with shorter-growing season environments requiring fast-maturity genotypes. This result may reflect the fact that longer growing season environments have higher rainfall, more frequent soil saturation, and/ or greater propensity for extreme rainfall events. In countries with higher annualised ratios of evapotranspiration to precipitation and lower risk of waterlogging (e.g., Australia), genotypes with waterlogging tolerance conferred relatively little benefit over the long-term.

**Future crop waterlogging stress patterns remain similar to those occurring historically**

We developed a new approach for clustering common stress patterns to facilitate functional insight into big data that would otherwise be outside the bounds of reasonable cognitive capacity. This characterisation of the timing of waterlogging stress as a function of phenology across diverse management, environments and climate types revealed two fundamental insights when assessed at the global scale. First, winter genotypes experience earlier seasonal patterns of waterlogging stress relative to spring genotypes (cf. Fig. 3a–f). Even though cereal crops are more sensitive to waterlogging during their reproductive phases, winter genotypes experienced greater yield penalty under early waterlogging (than spring genotypes under later waterlogging),

because waterlogged durations experienced by winter genotypes were generally longer (Figs. 3 and 4a, d). Second, even though future crop waterlogging events are likely to increase by 2–10% (Supplementary Fig. 7), we revealed the serendipitous outcome in which waterlogging stress patterns for each of winter and spring genotypes under present conditions are likely to be similar to those expected in future climate (Supplementary Fig. 8). Equipped with such knowledge, agronomists and crop breeders would likely achieve more widespread impact if new spring genotypes were adapted to late-season waterlogging, while proposed development of new winter barley genotypes would likely achieve wider impact if designed with early waterlogging in mind. It should be noted that while situations with minimal waterlogging stress (SW0 and WW0) would predominate (Fig. 3c, f); this result does not guarantee that such environments will not experience waterlogging stress under future climate, rather, that low waterlogging stress is more likely to emanate over the long-term[40,47].

Similar frequencies of waterlogging under historical and future conditions is a fortuitous outcome, because it suggests that practitioners could effectively develop today's adaptations for the temporal waterlogging patterns of tomorrow. If future waterlogging-stress patterns were dissimilar to those occurring historically, then the design of effective adaptations to future conditions would be more hindered due to the need to establish controlled-stress environments[48] or create synthetic waterlogging stress patterns similar to those expected in future. However, similar historical and future waterlogging stress patterns suggest that beneficial adaptations within each waterlogging stress pattern—e.g., the early-onset severe pattern of winter barley— could be readily transferred between regions, production systems and time periods, provided that other factors remained unchanged (e.g., local-adaptation of genotypes for disease resistance). Clustering stress patterns into common groups allows us to move away from locally specific factors causing the waterlogging stress (e.g., poor drainage, rising groundwater, superfluous rainfall, sowing time, genotype, soil type

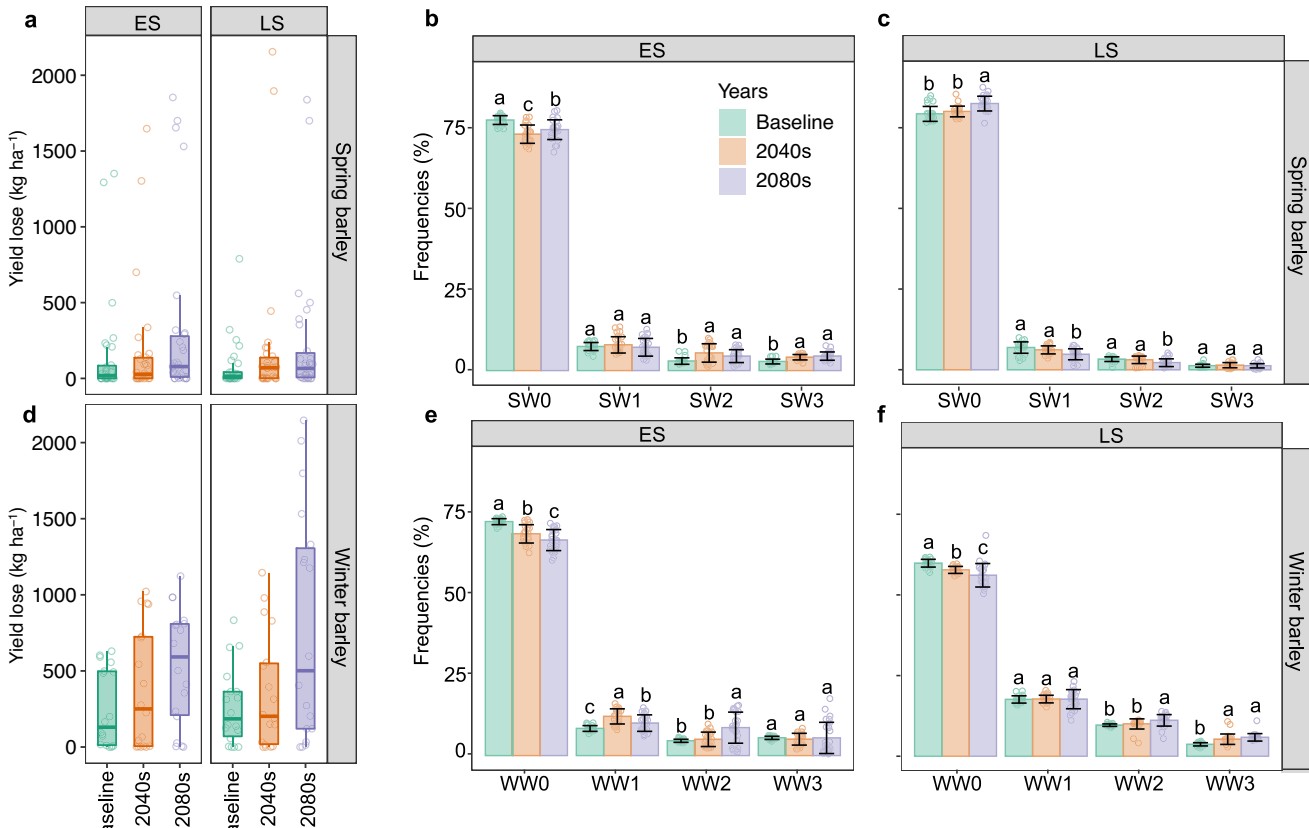

**Fig. 4 | Grain yield penalty and waterlogging stress patterns for the baseline (1985–2016), 2040 (2030–2059) and 2080 (2070–2099).** Grain yield penalties are shown for spring (**a**) and winter (**d**) barley across sites and genotypes for relatively early or late sowing (ES, LS) at each site. Boxplots indicate yield penalty for spring and winter barley across sites and GCMs ($n = 27$); box boundaries indicate 25th and 75th percentiles, whiskers below and above the box indicate the 10th and 90th percentiles, respectively. Waterlogging stress patterns for spring barley include SW0 (minimal waterlogging); SW1 (low moderate-late waterlogging); SW2 (late-onset moderate waterlogging); SW3 (late-onset severe waterlogging) and winter barley, WW0 (minimal waterlogging); WW1 (low early-onset waterlogging relieved later); WW2 (moderate early-onset waterlogging); WW3 (severe early-onset waterlogging). Data in (**b**), (**c**), (**e**) and (**f**) are presented as mean ± standard errors of the mean of GCMs ($n = 27$). Data were analysed using one-way analysis of variance followed by least significant difference (LSD) post-hoc tests. Different letters in (**b**), (**c**), (**e**) and (**f**) above the bars indicate significant differences in the frequency of stress patterns between climate periods ($P < 0.05$). Exact $P$ values include: 4.96e-08 (SW0 for ES in spring barley), 0.46 (SW1 for ES in spring barley), 1.47e-04 (SW2 for ES in spring barley), 2.61e-08 (SW3 for ES in spring barley), 7.44e-07 (SW0 for LS in spring barley), 2.88e-05(SW1 for LS in spring barley), 5.39e-04 (SW2 for LS in spring barley), 0.49 (SW3 for LS in spring barley), 1.42e−11 (WW0 for ES in winter barley), 2.30e-08 (WW1 for ES in winter barley), 5.66e-06 (WW2 for ES in winter barley), 0.84 (WW3 for ES in winter barley), 3.97e-07 (WW0 for LS in winter barley), 0.97 (WW1 for LS in winter barley), 4.93e-04 (WW2 for LS in winter barley) and 6.01e-09 (WW3 for LS in winter barley).

etc.) to the stress pattern that would most likely be realised in a given environment as a function of crop phenology.

## On the implications of regional climate change for waterlogging and grain yield

Our work has shown that mean grain yield penalty caused by waterlogging increased from 6 to 14% in 2040 to 10–20% by 2080 across GCMs, genotypes, management and sites. This result encompasses locally specific findings for Europe (e.g., France, Germany, UK and Spain)[49] and China[43] under superfluous precipitation scenarios. In these regions, yields were higher under future climate due to elevated atmospheric $CO_2$ concentrations and moderate alleviation of cold stress when water was not limiting[50–52], analogous to yield gains seen in US dairy systems[3,16]. Our findings also align with previous work which suggests that winter crop yields in Europe will rise by 2050[53] due to greater biomass production, grain number and grain weight associated with a fertilisation effect of atmospheric $CO_2$ and moderate warming[11]. In waterlogging-prone regions within Australia, we showed that yields are likely to increase under future climate due to a lower incidence of waterlogging (Fig. 2). Less rainfall in regions with high precipitation (>600 mm/year) may reduce disease susceptibility (e.g.,

stripe rust), improve crop health and further raise yield under future climate, although it should be noted that biotic pressures were not accounted for in the modelling framework used here (e.g. Snow et al.[54]). To avoid confounding impacts of superfluous water with those of nitrogen stress, we ran simulations without N stress invoked, albeit effects of waterlogging on mineral N and the corollary of such interplay would be a fruitful endeavor for future research (Rawnsley et al.[55]).

Climatic transition towards drier and hotter conditions by the end of 21st century is projected for many regions, often with an increased likelihood of extreme weather events[40,41,50,52]. Even though future climate were conducive to a 2–10% higher risk of severe waterlogging across the entire solution space (Supplementary Fig. 8), high variation between regions and genotypic lifecycles (Supplementary Figs. 3 and 8) was offset by the beneficial effects of climate change that collectively improved yield by 8–17% under future climate. As part of this, we found higher frequencies of early-onset severe waterlogging stress in Argentina, Ethiopia, China, the UK, France and Germany, in line with reports of increased flash flooding in some regions towards the end of the 21st century, particularly Asia and Africa[56]. We suggest that particular attention should be placed on the development of waterlogging

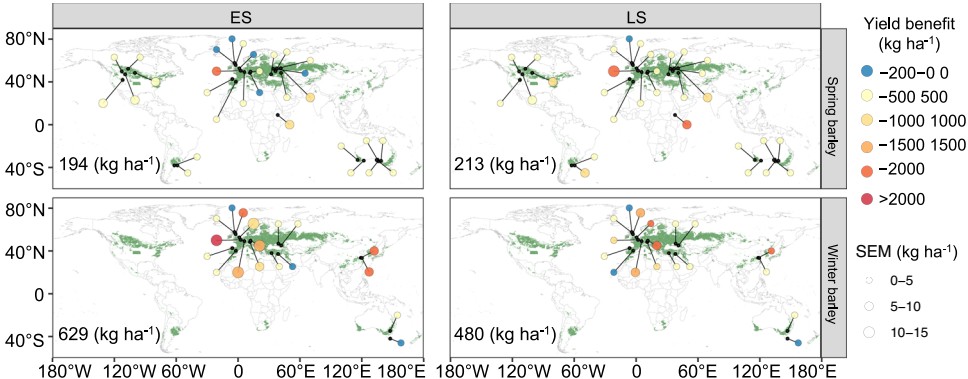

**Fig. 5 | Mean and standard error of the mean (SEM) for grain yield benefit associated with waterlogging tolerant genotypes relative to waterlogging susceptible genotypes for 2040 (2030–2059).** Values were computed across years and 27 GCMs in which barley growing season rainfall was higher than the 90th percentile; numerical values shown in each panel represent mean grain yield benefit across sites, years and GCMs. The map was modified using R package ggplot2'maps (version 3.4.0)' with the Natural Earth dataset in a public domain (https://www.naturalearthdata.com).

mitigation approaches for smallholders and the rural poor in lower-latitude countries where increased flood frequency is projected and prevailing rainfall is already high; women, youth and marginalised groups need to be empowered for proposed adaptation approaches to be successful. To engender adoption, appropriate research, development, policy and extension packages will be required to ensure that proposed adaptations are cost-effective, demand-driven, socially responsible and equitable[57,58].

### A requirement for contextualised adaptation to future climate change

The effectiveness of genotypic adaptation (i.e., introduction of crops with waterlogging tolerance genes) was higher in Ethiopia, China, Germany, France and UK. Across countries analysed, we showed that the adoption of waterlogging tolerant genotypes could mitigate up to 18% yield penalty caused by waterlogging under future climate, suggesting further research and development of such genotypes would be a worthwhile investment. In other regions, converting from longer-season winter genotypes to short-season spring genotypes could help avoid waterlogging, but with regional specificity viz. long-season waterlogging tolerant genotypes were shown to be more effective in Ethiopia, while short-season waterlogging tolerant genotypes were more effective in Europe and China. Taken together, our results suggest that contextualised adaptation will be key: there is no panacea, and certainly no singular generic solution for all environments. Fruitful future research may include 'stacking' or combining of several beneficial adaptations to determine whether the benefit from individual adaptations is synergistic or antagonistic[5,51].

As far as we are aware, the present study is the first experimental quantification of waterlogging expected under future climate in each of the major barley cropping zones of the world. To quantify waterlogging stress patterns, we develop and exemplify a simple, transferable approach for clustering crop stress patterns across regions, climate, management and genotypes. We show that even though the frequencies of global waterlogging will become higher, these changes will be outweighed overall by reduced waterlogging in other regions together with elevated $CO_2$ and warmer growing season temperatures. Our clustering approach comprises a pathway in which diverse bioclimatic applications may be able to categorise big data outputs into functional and biologically-meaningful patterns. While we apply this method to waterlogging and barley, although the framework could be readily applied to any crop, production system, or temporal biological variable. With regards to adaptation, we show that waterlogging tolerance genetics will have benefit in Ethiopia, France and China, but particularly in regions were long-season 'winter' genotypes are commonplace. Shifting from relatively late to early sowing or from late to

early maturity genotypes may alleviate waterlogging-induced yield penalties in some environments (Australia, Canada, Spain, Turkey and USA).

In this study, we used APSIM based on evidence that suggests that this farming systems framework is one of the most reliable models for simulating waterlogging dynamics[57,59]. Increasingly, however, multi-model ensemble studies for predicting agroecological variables are becoming commonplace, associated with the rise of high-performance computing, big data and cloud analytics[30,31,60]. Some ensemble studies suggest that taking either the ensemble mean or median of simulated values provide more accurate estimates than any individual model when variables related to growth are considered[61,62]. Indeed, the authors of the present study are working as part of an international research team in an Agricultural Modelling Intercomparison Project (AgMIP)[59] to test the applicability of our new approaches in a global study of crop waterlogging. This will allow us to scale our developments from barley to other genotypes, management options and environments using a range of models and, together with co-design as part of a community of AgMIP practitioners, improve the rigour of the approaches developed here.

While we only used one crop model, we invoked projections from an ensemble of outputs from 27 global climate models (GCMs). This aspect could be construed as both a strength and a weakness; the former because the ensemble mean of climatic projections should be more reliable than a projection from any one GCM (as discussed above), the latter because the variability in modelled outputs increases associated with greater variability in climatic realisations. Larger variability in outputs increases the uncertainty associated with the projection and can make results from such studies more difficult to comprehend in a rationally bounded way[50,63]. In fact, such diversity in potential simulated results across sites, seasons, genotypes and management was a key reason we developed the new approach to cluster waterlogging stress patterns.

Of all GCM outputs, rainfall is perhaps the most uncertain. A key reason for using the data from 27 GCMs was to better quantify the spatiotemporal distribution of and variability in precipitation during the growing season. We downscaled the GCM datasets using the NASA/POWER gridded historical weather database[64]. However, previous work has shown that interpolated gridded data tends to be conducive to producing rainfall events that are smaller in quantum but more frequent, which can lead to lower surface runoff and higher soil evaporation[64]. In a crop model, this could reduce plant water and nitrogen uptake, resulting in the propagation of errors that impact on variables such as biomass and yield. Using agricultural systems models with observed data before spatially interpolating point-based results may thus represent a more preferable approach for reducing

uncertainty in model outputs. While the present study avoids the aforementioned issue associated with nitrogen uptake because nitrogen stress was not invoked, in practice, mineral nitrogen deficiencies associated with waterlogging may be present because waterlogging impacts on the ability of plant roots to uptake nutrients[65,66].

Although we revealed multiple prospects for alleviating crop waterlogging under future climates, the variability in simulated yield and phenology responses under future climates highlights the importance of genotypic sensitivity to waterlogging stress. Across scenarios, mean yield penalty from waterlogging increased from 3–11% (baseline) to 6–14% (2040) and 10–20% (2080). Potential yield losses largely depend on genotypic sensitivity to waterlogging stress, in general with greater yield gains for tolerant genotypes of early sown (winter maturity) waterlogging tolerant genotypes, and the lowest gains for later sowing of (spring maturity) waterlogging tolerant genotypes (Fig. 5). We obtained genotypic parameters for waterlogging tolerance and phenology from previous empirical studies[40,67] but additional parameters from local genotypes would help improve the rigour of projected changes under future climates. However, we emphasise that the relative difference between scenarios is more important than the absolute values in this study.

## Methods

### Experimental data used for parameterisation and evaluation
Measured data from five two-year experiments (Exp1, Exp2, Exp3, Exp4, Exp5) conducted in five countries (Australia, Argentina, Canada, China and Ireland) were used for model development and evaluation. Exp1 was conducted under controlled conditions (Mt Pleasant Laboratories, Launceston, Tasmania, Australia) with four waterlogging treatments using six contemporary Australian barley genotypes differing in their waterlogging tolerance from 2019 to 2020 (see refs. [39,40]). In Exp2, barley yields were measured under five waterlogging treatments in the greenhouse and field conditions at the School of Agronomy, University of Buenos Aires, Argentina in 2010 (see ref. [68]). In Exp3, barley genotypes were evaluated for waterlogging tolerance in controlled field conditions at Brandon Research and Development Centre, Brandon, Manitoba, Canada from 2016 to 2017. Waterlogging treatments were initiated at tillering by adding the water to heights of 0.5–1 cm above the soil surface (see ref. [69]). In Exp4, barley yields were measured in field conditions carried out at Oak Park, Carlow, Ireland from 2017 to 2018. Waterlogging treatments were initiated at the tillering stage using a boom irrigator (see ref. [70]). In Exp5, field experiments were conducted in 2003–2004 and 2005–2006 at Zhejiang University, Hangzhou, China. Waterlogging treatments were imposed at tillering (see refs. [71,72]). All experiments were carefully managed to provide adequate nutrition and control of biotic pressures. Original datasets used for model evaluation (e.g., yield under ambient conditions and those subject to waterlogging) were compiled from a range of environments, including field experiments and environmentally controlled experiments. Given the diversity in data origins and variation in units used for reporting (e.g., g plant$^{-1}$, g m$^{-2}$, kg ha$^{-1}$), we standardised grain yield dimensions to kg ha$^{-1}$ before computing yield loss using Eq. (1):

$$Yield\ loss(\%) = \frac{(Yield_{ck} - Yield_{WL})}{Yield_{ck}} \times 100\% \qquad (1)$$

Where $Yield_{CK}$ is the yield (kg ha$^{-1}$) obtained from control treatment and $yield_{WL}$ represents yield measured for the waterlogging treatments (kg ha$^{-1}$).

### Advancing the process basis of APSIM-Barley for the simulation of waterlogging
We embedded the aforementioned experimental data into APSIM-Barley using physiological constructs detailed below to improve the ability of the model to simulate waterlogging[40]. The waterlogging tolerance genes examined here enable plants to tolerate saturated soils by accelerating aerenchyma formation and increasing root porosity following a waterlogging event. To account for this, we conceptualised three stages of plant response and adaptation (Supplementary Fig. 9). Stage one is the immediate plant response to waterlogging at which time water supply to the plant is unlimited and with soil strength lowered, root growth has little physical impediment. In this phase, biological functioning is not limited by oxygen or water availability and growth processes are not affected. During the second stage, soil water pores become fully saturated and oxygen-dependent bioprocesses are negatively influenced. The default version of APSIM contains photosynthesis functions for waterlogging (oxdef_photo), but these do not account for the effects of waterlogging on phenological rate. To compare the improved version of APSIM (detailed below) with the default version, we ran the improved model twice: once with oxdef_photo set to 1, and again with these parameters using values detailed as below. To account for genetic differences in waterlogging tolerance in APSIM, we developed and added the function *y_oxdef_lim_photo*, where values of unity or nil equate to no stress or full stress, respectively. The third stage encompasses adaptation responses, the net result of which is a variable level of adaptation depending on waterlogging tolerance genetics. After the adaptation stage, genotypes tolerant to waterlogging tend to exhibit similar photosynthetic rates compared with before waterlogging, analogous to plants that grow aerenchyma after waterlogging[39], whereas genotypes sensitive to waterlogging can exhibit decreased growth after waterlogging events if *y_oxdef_lim_photo* remains less than unity. We did not mathematically transcribe a process for crop failure under waterlogging, because in the majority of cases, waterlogging is realised as a transient event (viz. Fig. 3) and our experimental work suggests that intolerant genotypes persist for up to two months of waterlogging without failing[40]. These concepts were programmed into the source code of APSIM; the executable containing the modified source code and XML files are available online: https://github.com/KeLiu7/Waterlogging-Barley.

### Sites for factorial simulations
Soil waterlogging occurs when soils become saturated and plant roots cannot respire, while flooding refers to excessive surface water accumulation[65]. Waterlogging may be present without surface flooding. Waterlogging can be caused by extreme rainfall events, prolonged seasonal rainfall, poor soil hydraulic conductivity, lateral surface and/or groundwater flows, rising/perched water tables, improper irrigation or combinations of these factors[57]. Despite the diversity of ways in which waterlogging can occur, the result is oxygen levels in pore spaces that are insufficient for plant roots to adequately respire[73]. To account for soil and climatic variability across regions, simulations were conducted using sites across thirteen countries based on national barley production and planting area. In each country, simulated sites were prioritised based on dominant soil types in cropping zones from the Digital Soil Map of the World[74]. These representative sites (Supplementary Table 3 and Supplementary Fig. 10) where barley is grown[75] have documented reports of waterlogging[76]. Soil parameters at each site (soil texture, bulk density, pH and organic carbon content etc.) were obtained from the International Soil Reference and Information Centre[77].

### Model calibration and evaluation
In APSIM-Barley (version 7.9)[78], phenology is described in terms of thermal time accumulation using 11 crop stages and 10 phases (time between stages). Further model details, including phenology and growth are detailed in refs. [78,79]. Site-specific genotype selection, crop management (e.g., sowing date) were based on local expert advice and experimental records (Supplementary Table 3). Genotypes were created such that lifecycles were in line with local sowing, flowering and

maturity times[80]. This was conducted by setting APSIM phenological parameters for vernalisation (*vern_sens*), photoperiod (*photop_sens*) and thermal time between emergence and the end of the juvenile phase (*tt_end_of_juvenile*). These parameters were chosen due to their high influence on crop flowering times[47]. Winter barley requires greater exposure to cold temperature to evoke reproductive development, whereas spring barley flowers without a cold exposure precondition. In APSIM-Barley, *vern_sens* refers to vernalisation representing cumulative cold temperature requirement to initiate reproductive development (range 0–5), while *photop_sens* refers to day length sensitivity (range 0–5); higher values denote greater sensitivity. We assigned *vern_sens* values based on maturity group ('spring' maturity = 1 and 'winter' maturity values of either 2.5 or 4, depending on vernalisation requirement)[81]. Similarly, *photop_sens* values were set to 1 for 'spring' maturity and 'winter' maturity values ranged from 2.5 to 4. tt_end_of juvenile values were set to 400 for 'spring' maturity and 'winter' maturity values ranged from 400 to 750. We first parameterised vern_sens according to the maturity group, we then adjusted photop_sens and tt_end_of juvenile until the simulated flowering days match with local flowering and maturity days.

Waterlogging results in the inhibition of processes in the mesophyll, photoassimilate transport in the phloem, gas conductance and thus reduces photosynthetic rate[57]. In APSIM-Barley, these processes are modelled per unit ground area. Effects of waterlogging on photosynthesis and phenology ('waterlogging-stress days') were modelled using stress indices (*oxdef_photo* and *oxdef_pheno*) computed as a function of the fraction of roots waterlogged (*oxdef_photo_rtfr*). For *oxdef_photo_rtfr* levels of 0.8 and greater, *oxdef_photo* and *oxdef_pheno* linearly decreased; for *oxdef_photo_rtfr* levels less than 0.8, no stress was invoked, following experimental observations[39,82]. Photosynthetic and phenological stress indices were defined as a function of crop stage (*x_oxdef_stage_photo, x_oxdef_stage_pheno*), which is a significant advance on the majority of previous studies which assume that waterlogging stress depends only on the extent and duration of water-filled pore space[67]. Part of the novelty of the current work is the delay in phenology associated with the duration of waterlogging and the crop stage/s in which it occurs. Waterlogging in early growth stages inhibits leaf appearance rate and tiller development and delays flowering. If waterlogging stress occurs during vegetative stages, plants may fully recover by grain-filling stages; if waterlogging occurs during flowering, plants cannot fully recover pre-waterlogging photosynthetic potential before maturity[40]. Effects of waterlogging on phenology (*oxdef_pheno*) were derived using information from environment-controlled experiments[40]. The parameter *oxdef_pheno* was computed as a function of the fraction of roots waterlogged (*oxdef_pheno_rtfr*). For *oxdef_pheno_rtfr* levels of 0.8 or greater, *oxdef_pheno* linearly decreased to 0.8 until the soil is fully saturated; for *oxdef_pheno_rtfr* levels less than 0.8, no stress was invoked.

To account for varying effects on phenology, we invoke the function *y_oxdef_lim_pheno* that is calculated according to crop stage (*x_oxdef_stage_pheno*, i.e., APSIM stage code). The *y_oxdef_lim_pheno* response function was adopted from our previous studies[40]. For *y_oxdef_lim_pheno* levels less than 1, crop phenology is delayed, with *y_oxdef_lim_pheno* increasing from 0.65 at stage 4.0 to 0.95 to APSIM stage 5.5; for *y_oxdef_lim_pheno* levels greater than 1, grain-filling durations are truncated, with *y_oxdef_lim_pheno* increasing from 1.0 to 1.5 between ASPIM stages 6 and 10. The delayed effect on phenology is only triggered before flowering (i.e., *x_oxdef_stage_pheno* between 1 and 6) and the grain filling duration reduction is triggered after flowering (i.e., *x_oxdef_stage_pheno* greater than or equal to 6). In general, the magnitude of delay is largely depended on the extent and duration of waterlogging stress, as well as its timing relative to crop development. The physiological basis for waterlogging-induced delays to phenology is discussed in our previous studies[40], with the rate of leaf emergence determining the duration between emergence and

anthesis[83]. Waterlogging in later growth stages causes premature flag leaf senescence and shortens the grain-filling period[66]. Reduced grain growth in waterlogged plants is due to decreased post-anthesis carbon assimilation and culm reserves remobilised to grains[84,85].

APSIM was initialised using the SWIM3 Module (soil water infiltration and movement; Supplementary Fig. 11). To examine the extent with which the new processes described above improved the ability to simulate crop growth and development under waterlogging, we also run a default (unimproved) version of APSIM-Barley with waterlogging. Simultaneous multi-objective optimisation[63] of *oxdef_photo* and *oxdef_pheno* for each genotype was performed for waterlogging treatments by minimising the sum of squared residuals across datasets. Evaluation of phenology, biomass and yield components under waterlogging have been described in our previous peer-reviewed literature[35,39,40]. Details of the waterlogging algorithms, including their implementation in the APSIM source code, are provided in peer-reviewed literature[40] and online: https://github.com/KeLiu7/Waterlogging-Barley.

### Historical and future climate data

Daily data for maximum and minimum temperature, rainfall and solar radiation for 1985–2016 at each location were obtained from the National Aeronautics and Space Administration/Prediction of Worldwide Energy Resources (NASA/POWER)[86]. NASA/POWER provides climate data at a horizontal resolution of 1° latitude–longitude. Yearly atmospheric $CO_2$ concentration [$CO_2$] for future periods were calculated based on the Shared Socio-economic Pathway 585 (SSP585), a business-as-usual (high) emission scenario. This scenario most closely represents the climate trajectory to date[41]. Yearly atmospheric concentrations [$CO_2$] were calculated for each year following the method[87] that used empirical equations obtained by nonlinear least-squares regression fitted to [$CO_2$] from the 27 GCMs (see Supplementary Methods).

$$[CO_2]_{SSP585} = 757.44 + \frac{84.938 - 1.537*y}{2.2011 - 3.8289*y^{-0.45242}} + 2.4712*10^{-4}*(y+15)^2$$
$$+ 1.9299*10^{-5}*(y-1937)*10^{-5}*(y-1937)^3$$
$$+ 5.1137*10^{-7}*(y-1910)^4$$

(2)

where *y* was the calendar year from 1900 to 2100 (year = 1900, 1901, …, 2100).

To generate climate scenarios for 2040 and 2080, monthly temperature, rainfall and radiation projected from 27 GCMs (Supplementary Table 2) are available from the Coupled Model Intercomparison Project Phase 6 (CMIP6). Here we used the statistical downscaling model (NWAI-WG)[88] to downscale GCM monthly gridded data to daily climate data for each of the study sites. Spatial downscaling was achieved by using an inverse distance-weighted (IDW) interpolation method in this study. The IDW interpolation method was used to compute rainfall and temperature values for each weather station based on its distance to the geographical centres of the four nearest GCM grid cells[89], then applied bias correction, resulting in bias-corrected monthly data using a relationship derived from observations and GCM data for the historical training period of 1985–2016. Bias-corrected and downscaled GCM trends were then transcribed into time series of daily maximum and minimum temperature, rainfall and radiation using a modified stochastic weather generator. The major advantage of this statistical downscaling method, particularly in comparison with more computationally demanding dynamical downscaling, is that it can be easily applied to any location for which a long-term daily historical climate record is available. We did not use daily data from GCMs in this study for three reasons: first, not all GCMs provided daily

climate data. Second, interpolation of daily GCM values can be error-prone, especially for rainfall. Third, weather variables such as radiation, precipitation, minimum and maximum temperature are often interdependent (e.g., rainy days are often cooler and have lower solar radiation); interpolation and bias-correction on a daily time-step can confound such interdependence[88]. Instead, the approach we used (NWAI-WG bias correction of monthly values) accounts for interdependency between climatic variables[89].

### Factorial simulations

Simulations were run from 1985 to 2100; initial soil conditions were reset annually at sowing to prevent potential 'carry-over' effects from previous seasons. Initial plant available water at sowing was set 15 mm to ensure consistency of emergence across sites and sowing dates. Barley was sown at 180 plants m² using a depth of 20 mm and row spacing of 200 mm. Nitrogen was applied as $NO_3^-$ and maintained above 200 kg ha⁻¹ in the top 300 mm throughout the season to ensure that nitrogen supply did not limit growth. This assumption was made so that waterlogging stress typologies described below were not confounded by the presence or absence of N stress. Soil texture, bulk density, pH, and organic carbon content were obtained from the International Soil Reference and Information Centre[77]. Global groundwater table depths used in model initialisation were obtained from Aquaknow[89].

### Novel approach for clustering seasonal waterlogging-stress typologies

To categorise waterlogging stress patterns, we output seasonal time courses of waterlogging stress on photosynthesis as a function of phenology (APSIM output variable *oxdef photo*). Individual stresses were clustered across simulation years, sites, genotypes and management. For each environment, waterlogged days (i.e., days with *oxdef photo* lower than 1) were cumulated for each of six discrete growth stages (i.e., early juvenile (JV1, 10 <= APSIM growth stage <21); late juvenile (JV2, 21 <= APSIM growth stage <32); floral initiation to heading (FIN, 32 <= APSIM growth stage <65); flowering to grain filling (FIN, 65 <= APSIM growth stage <71); early grain filling (GF1, 71 <= APSIM growth stage <80); late grain filling (GF2, 80 <= APSIM growth stage <87). *oxdef photo* was averaged for each growth stage across simulation years, sites, genotypes and management. Prevailing seasonal waterlogging patterns were realised by applying unsupervised *k*-means clustering to all seasonal trajectories of *oxdef photo* against phenology. Clustering was applied using the R statistical package 'stats' (R Development Core Team, 2013), with clusters being defined such that total within-cluster variation was minimised (partitioning *n* observations into *k* clusters (the value of K is assigned as four) where each observation belongs to the cluster with the nearest mean, i.e., the cluster centroid).

### Simulated impacts of waterlogging on yield

The impact of waterlogging stress on crop yield for a given location was quantified by comparing the yield difference of each year simulated by the default version of APSIM and improved APSIM with waterlogging algorithms. The yield difference caused by waterlogging ($Yield\ percentage_{WL}$) was calculated using Eq. (3):

$$Yield\ percentage_{WL} = \frac{Yield_y - Yield_{y,wl}}{Yield_y} \times 100\% \qquad (3)$$

Where $Yield_y$ represents simulated grain yield (kg ha⁻¹) obtained using a default APSIM version with a climate period from 1985–2100 (Y = 1985, 1986, ..., 2100) and $yield_{WL,y}$ represents simulated yield (kg ha⁻¹) obtained from the modified version of APSIM using the aforementioned waterlogging algorithms for the said year.

### Reporting summary

Further information on research design is available in the Nature Portfolio Reporting Summary linked to this article.

## Data availability

The simulated yield data generated in this study are provided in the Source data file. Genotypic parameters used in APSIM are available in Supplementary Table 3. Soil data and downscaled climate datasets are available online (https://doi.org/10.5281/zenodo.7444483)[90]. Source data are provided with this paper.

## Code availability

The R code containing the clustering algorithm and the APSIM executable containing the improved waterlogging algorithms are available in the GitHub repository (https://doi.org/10.5281/zenodo.7444483)[90].

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

## Acknowledgements

The research was financially supported by Grains Research and Development Corporation grant (UOT1906-002RTX) issued to M.T.H. We are grateful to Isaiah Huber from Iowa University for APSIM programming.

## Author contributions

K.L. and M.T.H. conceived the study. K.L., H.L.Y. and D.L.L. conducted the crop model simulations and downscaled global climate models. K.L., M.T.H. and S.A. coded the waterlogging functions into APSIM source code. K.L., M.T.H. and H.L.Y. created and analysed the results. K.L. and M.T.H. wrote the paper. M.T.H. funded the study. All authors edited and revised the paper.

## Competing interests

The authors declare no competing interests.

## Additional information

[1]Tasmanian Institute of Agriculture, University of Tasmania, Newnham Drive, Launceston, TAS, Australia. [2]MARA Key Laboratory of Sustainable Crop Production in the Middle Reaches of the Yangtze River (Co-construction by Ministry and Province), College of Agriculture, Yangtze University, Jingzhou, China. [3]State Key Laboratory of Cotton Biology, Institute of Cotton Research of the Chinese Academy of Agricultural Sciences, Anyang, China. [4]New South Wales Department of Primary Industries, Wagga Wagga Agricultural Institute, Wagga Wagga, NSW, Australia. [5]Climate Change Research Centre, University of New South Wales, Sydney, NSW, Australia. [6]Department of Agricultural and Biological Engineering, IFAS, University of Florida, Gainesville, FL, USA. [7]Agroecosystem Sustainability Center, Institute for Sustainability, Energy, and Environment, University of Illinois at Urbana Champaign, Urbana, IL, USA. [8]College of Agricultural, Consumer and Environmental Sciences, University of Illinois at Urbana Champaign, Urbana, IL, USA. [9]National Center for Supercomputing Applications, University of Illinois at Urbana Champaign, Urbana, IL, USA. [10]NASA Goddard Institute for Space Studies, New York, NY, USA. [11]Columbia University, Center for Climate Systems Research, New York, NY, USA. [12]Potsdam Institute for Climate Impacts Research (PIK), Member of the Leibniz Association, Potsdam, Germany. [13]Commonwealth Scientific and Industrial Research Organisation (CSIRO) Agriculture and Food, Canberra, ACT, Australia. [14]State Key Laboratory of Grassland Agro-ecosystems, College of Ecology, Lanzhou University, Lanzhou, China. [15]College of Agronomy and Biotechnology, China Agricultural University, Beijing, China. [16]Department of Agronomy, Iowa State University, Ames, IA, USA. [17]Research Center for Physiology and Ecology and Green Cultivation of Tropical Crops, College of Tropical Crops, Hainan University, Haikou, Hainan, China. [18]Brandon Research and Development Centre, Agriculture and Agri-Food Canada, 2701 Grand Valley Road, Brandon, MB R7A 5Y3, Canada. [19]MARA Key Laboratory of Crop Ecophysiology and Farming System in the Middle Reaches of the Yangtze River, College of Plant Science and Technology, Huazhong Agricultural University, Wuhan, Hubei, China. [20]National Institute for Agricultural Research (INRAE), UMR AGIR, Castanet Tolosan, France. [21]College of Resources and Environmental Sciences, China Agricultural University, Beijing, China. [22]Department of Agronomy, Abdul Wali Khan University Mardan, Khyber Pakhtunkhwa, Pakistan. [23]Joint International Research Laboratory of Agriculture and Agri-Product Safety of the Ministry of Education of China, Yangzhou University, Yangzhou, Jiangsu, China. [24]Key Laboratory of Land Surface Pattern and Simulation, Institute of Geographic Sciences and Natural Resources Research, Chinese Academy of Sciences, Beijing, China. [25]Natural Resources Institute Finland (Luke), Latokartanonkaari 9, 00790 Helsinki, Finland. [26]School of National Safety and Emergency Management, Beijing Normal University, Beijing, China. [27]University of Göttingen, Tropical Plant Production and Agricultural Systems Modelling (TROPAGS), Grisebachstr. 6, 37077 Göttingen, Germany. [28]School of Computer Science & Information Engineering, Anyang Institute of Technology, Anyang, China. [29]These authors contributed equally: Ke Liu, Matthew Tom Harrison, Haoliang Yan. ✉ e-mail: matthew.harrison@utas.edu.au

