## [Peer Review File · Nature Communications]

Reviewers' Comments:

Reviewer #1:

Remarks to the Author:

The manuscript is very well conceived taking a broad spectrum of original data into the model studies.

The differences between the building of the different crop models needs more detailed explanation and the step to step changes of the various models.

Some more detail how data from different published studies were 'standardized' to use for this modelling work is required. since I presume original input data varied greatly an explanation needs to be added on how the authors dealt with missing information.

The data availability and code availability are insufficiently enabled. The definition of 'upon reasonable request' needs to be reconsidered, links to data depositories need to be provided for simulated yield data. The R code needs to be deposited in a public depository.

Reviewer #2:

Remarks to the Author:

General comments

Soil waterlogging is increasingly a global problem due to increased frequencies of extreme climate events. However, less attention has been paid to waterlogging compared with heat and drought stress. The authors did a great job in simulating the effect of waterlogging stress on barley production across global environments using the improved APSIM model. The method looks robust, and the findings are of importance to a wider community. Generally, the work is well organized, and the readability of the manuscript is also good. Several issues should be solved before going further.

Model performance.

1. The process basis for simulating waterlogging in the APSIM model shows significant advantages over many other crop models without the inclusion of soil waterlogging. It would be nice to discuss the model performance using available observed datasets. This is because yield reduction caused by waterlogging stress is generally from multiple aspects, e.g. delayed phenology (the improved APSIM shows this function), reduction in biomass, grain number etc.

2. The evaluation of soil moisture dynamics and water table depth in the model, which is the basis for simulating waterlogging stress on crop growth. I can only see some simulated and observed results relevant to water table depth in the supplementary information. How about the model's performance on soil water dynamics?

3. Soil waterlogging may occur for many reasons, including extreme rainfall events, intense rainfall in short periods, poor hydraulic conductivity, rising/perched water tables, or combinations of these factors. I also checked your previous publication (Liu et al., 2020 FES) that calls for a better understanding of the model's sensitivity to hydraulic conductivity. Here you have examined 38 sites that have different soil properties in your study. How the sensitivity of those factors to waterlogging occurrence was addressed here?

Modelling waterlogging process

4. Your previous paper (Liu et al., 2020) suggests that the watertable depth seems never exceeded the plant height. Heavy rainfall could cause inundation and submergence, which could cause crop failure. Does the APSIM model account for crop failure due to submergence?

5. How are waterlogging-stress days defined in the improved model? and how the four waterlogging types were identified here? The information in Fig. 3 is not very clear to me.

Results

6. Schematic of genotypic traits influenced by waterlogging in Fig. 1 is not intuitive. Please provide more descriptions of how those functions work in the model.

On the other hand, the simulated results in the improved APSIM were much better compared with the default APSIM model. I'm wondering if the waterlogging functions in the improved model could emerge in other crop models, which will benefit the crop modelling community a lot. Please note this is not a real remark, but a point of potential discussion.

7. Based on Fig. 2, it seems that winter barley has a greater yield penalty by waterlogging stress. I am sure that you are talking about sowing time here. Spring sowing has much less chance of

waterlogging thus less difference between different models. As in Sup Table 2, spring barley refers to spring type barley and winter barley refers to winter type barley, the terms "spring sowing barley" and "autumn/winter sowing barley" should be used here.

8. The typology in Fig. 3 is another main result of the paper. You seem to refer to each class of waterlogging as a "typology", which might be not very accurate. What's the basis to support this statement? The abbreviation for WL is not explained in the caption.

9. Early sowing and late sowing should have consistent abbreviations. In Fig. 2, it was E and L, respectively and it was ES and LS in Fig. 4 (typos in the caption of Sup Fig 3). Also, please clarify the significant level, at 0.05 or 0.01.

10. In Fig. 5, you said that altering sowing time coupled with the adoption of superior genetics resulted in further gains in yield. Why the results (yield benefit) are negative in some places? Also, is this correct that SEM was so small?

Discussion

11. Your results showed that the overall mean yield penalty from waterlogging increased from 3-11% (baseline conditions) to 6-14% (2040) and 10-20% (2080). The yield loss largely depends on the genotypes' sensitivity to waterlogging stress thus genotypic effects should be discussed (i.e. potentially under-estimated or over-estimated).

12. Authors should also discuss the limitation of this study. For example, only one model (APSIM) was used in this study. Martre et al. (2015) indicated that multi-model ensembles are much better than one. The limitations of multi-model ensembles and the possibility of transferring the waterlogging functions developed in APSIM to other models should be discussed.

13. At the end of the discussion, a paragraph explaining the uncertainty mentioned above of the research should be added.

Materials and method

14. Authors collected some yield datasets of waterlogging experiments globally to evaluate the model performance. How do the authors collect the soil and climate datasets from each location? Please correct the title as "Data for waterlogging function parameterisation and validation".

15. Authors proposed a good idea on crop response, adaptation and recovery from waterlogging for alternative genotypes. However, it is still confusing how the waterlogging-tolerant genotypes were simulated in the model. Are there any parameters to account for this? if yes, please provide more details in the "Impact of waterlogging tolerance genes on barley growth and development."

16. For future climate scenarios, could you please explain why did you use monthly data instead of daily data at the first step (assuming that daily data are available for some of the mentioned GCM)?

17. This is a projection study. Please explain why the simulation climate periods of 2030-2059 and 2070-2099 were chosen.

18. Please state what kind of R packages were used to do the K-means cluster in this study.

20. Add some explanation of the formula that how the atmospheric CO₂ concentrations were calculated.

21. I noticed that you compared the yield difference between the improved model and the default model. Did you run the model two times or set the waterlogging parameters at 0 to remove waterlogging effects? Please indicate if the default APSIM model includes waterlogging or not. This is important to evaluate the waterlogging effects on crop production.

22. It's okay to create hypothetical genotypes with factorial variations in traits to represent the local cultivars. However, not all modellers are familiar with APSIM model, thus more details on the basis for setting the parameter for spring and winter barley, respectively, would be helpful.

Some other minor comments:

Page 4, Y or y in the function should be consistent.

Page 20, please provide the full name for AgMIP.

Supplementary Table 1. The full name of GCM should be global circulation models.

Supplementary Table 3, please use color blind friendly to make your Table accessible.

Reviewer #3:

Remarks to the Author:

General comment

This paper demonstrated the potential of crop models for decision-making in an agronomy field un-explored such as crop waterlogging stress. It is original and it has great significance for the agronomy and food security fields. The work meets the expected standards in the crop modelling field except for one reason, which is crucial for model developers, it has a lack of code review by independent peers. Therefore, the main limitation of this work is that the presented results can have a bias related to the updated/modified Barley code which was not reviewed by independent researchers from the crop modelling community and property released (or at least it was not stated in the manuscript). I strongly suggest to update the code review status or wait until the updated Barley model is properly reviewed and released by the APSIM Initiative Panel before this paper can be published.

Specific comments

L121. Is there an appropriated target stress pattern? Generally, the target is to avoid the stress to increase yield production. It is difficult to think that a stress pattern can be targeted.

L442. Although the magnitude of the simulation analysis was done by the author is massive, a proper review of the updated model code is required before publication. If the updated Barley APSIM model was already reviewed by peers and released in the APSIM GitHub repository, please share the link to the code in this manuscript. Otherwise, the authors should wait to get the code reviewed by an independent group of researchers/programmers before publication of results using the updated model. As I understood an APSIM crop model (such as the updated Barley model presented here) is a prototype if it was not reviewed and approved by the APSIM Initiative Panel.

L449. Why did you use only soil and not climate to prioritized sites?

L478. The authors name APSIM version 7.9 to explain the phenology module in the Barley model. However, they did not explicitly name the model version used for this study which makes difficult to check code availability in GitHub. Although a GitHub repo was shared (<https://github.com/KeLiu7/Waterlogging-Barley>) it does not show the updated APSIM Barley model. The code shared is not providing any information about the updates implemented in the model.

L483. Which method did you use for genotype parametrization in the model? The authors stated they 'match' the thermal time between emergence and maturity for each site by adjusting phenological parameters. However, the model bias (observed - predicted) can be generated for any of these parameters. How you know the contribution of each parameter to the total model bias?

L524. Accordingly with the authors a reset function was not applied every year, therefore there is sequence/rotational effect, i.e., the water/N and C scenario at the end of a crop affects the water/N and C dynamics for the following crop. How did you count for the effect of rotation in your model simulations?

L525. In some regions of the world (e.g., NSW Australia), sowing date is defined by soil moisture content. During dry years, it is expected to have less sown area or in some cases crop failure due to lack of water availability for emergence. APSIM can model this behaviour. Why did you not apply variable sowing date based on water availability? This will produce more realistic long-term yield predictions based on interannual rainfall variability under future climates.

Reviewer #4:

Remarks to the Author:

The current study quantifies the effects of waterlogging on barley at different sites across the globe by developing a new modeling routine. In addition, they tested the potential of waterlogging tolerant cultivars with early and late maturity characteristics combined with shifting the sowing dates of spring and winter barley as adaptation strategies for climate change using the adjusted waterlogging module. The authors suggested that the developed pipeline can also be applied to

other crops and environments. They projected yield reduction due to waterlogging would be between 10% and 20% by 2080. However, the CO₂ fertilization effects can overcompensate for the adverse impacts of climate change with or without considering waterlogging response in the model for most of the study locations. Shifting the sowing date together using resilient cultivars would significantly decline the negative effects of waterlogging. There are relevant research questions as well as an essential topic addressed in this manuscript. Recent studies such as Webber et al., 2020 indicated the importance of more robust consideration of waterlogging in process-based crop models employed for impact assessment studies. Therefore, the significance of the research is undoubtedly, but the methodology is not novel. Excellent writing is evident in the manuscript. Aside from that, the presenting items are clear and informative. However, some issues regarding the model development, mechanism understanding, and assumptions for future projections need to be addressed before publishing as:

- Model development: It is unclear (or maybe I did not get it!) how and based on which physiological mechanisms the waterlogging effects on phenology were implemented in the model. (a) Is the timing of the specific phenological stages advanced or delayed under waterlogging (written in the methods section)? If yes, what are the physiological bases for it? The authors referred to their published studies (Liu et al., 2020 (discussed in section 4.3); Liu et al., 2021) as a base for the new modeling routine. However, I did not find a concrete physiological base for phenology response in that study that can implement in a process-based model. Climate change can advance the timing of sensitive periods such as flowering, which can alter the overlap between waterlogging period and those phenological stages, but it has nothing to do with the direct response of phenology to waterlogging.

(b) Whether different sensitivity to waterlogging is implemented depending on the phenological stage (that is, the sense getting from the main text)? This case is not new and was available on the old codes of APSIM. As far as I remember, there was an aeration deficit factor depending on the phenological stage in APSIM, which can linearly reduce the plant growth rate under waterlogging (high sensitivity in early growth stages and lower on maturity). Please check Asseng et al., 1997-figure 1.

Please indicate how exactly phenology considers in the new modeling routine, including the physiological mechanism behind the crop response, to make it clearer to readers from the current manuscript without the need to read at least two other manuscripts to understand the methodology.

What about photosynthesis's response to waterlogging? In figure 1, the authors mentioned radiation use efficiency (RUE) and photosynthesis in parenthesis. Those of two different modeling terminologies for converting intercepted radiation to biomass. Please be very specific about what exactly influences by waterlogging in the new routine. A phenological specific reduction factor on RUE or photosynthesis? It also surprised me why the authors did not mention the effects of waterlogging on transpiration (due to stomatal closure) as the most commonly known crop response to waterlogging in other models. Do you have such a response in your APSIM version? Modeling of waterlogging is not well developed, such as drought modeling (it has some reasons I explain below), but we have well-tested robust routines in other models, such as DRAINMOD (Skaggs et al., 2012) and SWAGMAN Destiny (Yang et al., 2016) which consider not only photosynthesis response but also transpiration and leaf area expansion. Why do we need to develop a new routine with fewer processes to consider?

The modeling of waterlogging is less developed because of the complexity of driving factors. To accurately simulate the impacts of waterlogging, local scale heterogeneity in topography, soils, severe compaction below the plough layer, and functional drainage must be taken into account, which can influence soil workability, crop establishment, and even nutrient leaching due to ponding versus runoff. How did APSIM consider those factors in a global analysis?

- Model calibration and validation: The parametrization and testing of the model are explained in a relatively general way, making it challenging to review the reliability of those processes. The authors mentioned in the text they only have phenology data for one experiment (line 551) and used only yield for parametrization of other experiments. This would substantially increase the risk

of getting the right results for the wrong reasons. The model development and phenology response are the core of current research therefore, the parametrization for phenology should be carefully treated.

- Limitations: The limitation of the current study needs to be clearly discussed. I suggest classifying them in two directions as input uncertainties and limitations in crop processes. The future climate projections are extremely uncertain regarding the temporal distribution of precipitation during the growing season, which is fundamental to assessing waterlogging risk therefore, we need to be careful in concluding waterlogging intensity for future windows such as the 2080s. The crop processes, such as early acclimation to waterlogging (as would be the case for winter barley in figure 3 d-f) (Herzog et al., 2015) or an increase in assimilate remobilization (Li et al., 2013) due to waterlogging not implemented in the model, would significantly change the results. The nitrogen stress was also switched off in the model execution however, it is against the nature of waterlogging since the roots under stress lost their nutrient uptake functionality. Please mention such limitations in discussing the results.

Minor issues:

- Please carefully define the difference between waterlogging and flooding in the text.
- Did you only consider rainfed systems or you had irrigated barley as well?

References

- Webber et al., 2020. No perfect storm for crop yield failure in Germany. *Environmental Research Letters*, 15: 104012.
- Asseng et al., 1997. Simulation of perched water-tables in a duplex soil. *Proceedings of MODSIM '97, International Congress on Modelling and Simulation, Hobart, Tasmania, Australia, 8-11 December 1997.*
- Skaggs et al., 2012. Drainmod: model use, calibration, and validation. *Transactions of the ASABE.*
- Yang et al., 2016. Prediction of salt transport in different soil textures under drip irrigation in an arid zone using the SWAGMAN Destiny model. *Soil Research*, 54(7): 869-879.
- Liu et al., 2020. Genetic factors increasing barley grain yields under soil waterlogging. *Food and Energy Security* 2020, 9(4): e238.
- Liu et al., 2021. Climate change shifts forward flowering and reduces crop waterlogging stress. *Environmental Research Letters*, 16(9): 094017.
- Herzog et al., 2015. Mechanisms of waterlogging tolerance in wheat – a review of root and shoot physiology. *Plant, Cell & Environment*, 39: 1068-1086.
- Li et al., 2013. Carbohydrates Accumulation and Remobilization in Wheat Plants as Influenced by Combined Waterlogging and Shading Stress During Grain Filling. *Journal of Agronomy and Crop Science*, 199: 38-48.

Reviewer #1 (Remarks to the Author):

The manuscript is very well conceived taking a broad spectrum of original data into the model studies.

Response: Thank you for your positive feedback.

The differences between the building of the different crop models needs more detailed explanation and the step to step changes of the various models. Some more detail how data from different published studies were 'standardized' to use for this modelling work is required. since I presume original input data varied greatly an explanation needs to be added on how the authors dealt with missing information.

Response: These are good suggestions. We added the following text with regards to the approach used to conceive, conduct and refine the study (see 'Conceptualising impacts of waterlogging on phenology and photosynthesis' and methods sections in the revised manuscript):

"We developed new functions to account for experimentally observed effects of waterlogging on photosynthesis and phenology (oxdef photo and oxdef pheno, respectively; Fig. 1a)³⁸. Each dimensionless function assumes multipliers ranging from unity to nil in the form of $y = f(x)$, where y is the stress factor and x is soil moisture. When x is at or below field capacity, $y = 1$; y linearly decreases with increasing x until the point at which the soil is saturated ($y = 0$).

Line 145-149

In concert with the original text below (square brackets), we feel that this description clearly articulates the approaches we used. We detail these approaches further in the methods.

[These functions were incorporated into the APSIM software platform to enable improved simulation of crop responses to waterlogging as part of an integrated system. We calibrated the waterlogging-enabled framework using published data from field observations across five countries (Australia, Argentina, China, Canada and Ireland; Supplementary Table 4)]

Line 149-153

We clarified the reviewer's query regarding 'standardization' in the methods as follows:

Original datasets used for model evaluation (e.g. yield under ambient conditions and those subject to waterlogging) were compiled from a range of environments, including field experiments and environmentally-controlled experiments. Given this diversity in data origins and existential variation in units used for reporting (e.g. g plant^{-1} , g m^{-2} , kg ha^{-1}), we standardised all dimensions for grain yield to kg ha^{-1} . Yield loss was then computed using Equation 1.

$$\text{Yield loss (\%)} = (\text{Yield}_{\text{CK}} - \text{Yield}_{\text{WL}}) / \text{Yield}_{\text{CK}} \times 100\% \quad (1)$$

Where Yield_{CK} is the yield (kg ha^{-1}) obtained from control treatment, while Yield_{WL} represents yield measured for the waterlogging treatments (kg ha^{-1}).

Line 466-473

Editorial note: Throughout the paper we have adopted Australian English (e.g. 'standardisation' instead of 'standardization'); however, we are happy for this to be altered in line with editorial guidelines for *Nature* journals.

The data availability and code availability are insufficiently enabled. The definition of 'upon reasonable request' needs to be reconsidered, links to data depositories need to be provided for simulated yield data. The R code needs to be deposited in a public depository.

Response: We revised this aspect in line with a request from the handling Editor (Dr Pilar Morera Margarit):

Data availability. Simulated yield data are available in Supplementary Data 1 and genotypic parameters used in APSIM are available in Supplementary Table 2. Soil data and downscaled climate datasets are available online: <https://github.com/KeLiu7/Waterlogging-Barley>

Code availability. The R code containing the clustering algorithm and the APSIM executable containing the improved waterlogging algorithms are available in the GitHub repository: <https://github.com/KeLiu7/Waterlogging-Barley>

Line 656-662

Reviewer #2 (Remarks to the Author):

General comments

Soil waterlogging is increasingly a global problem due to increased frequencies of extreme climate events. However, less attention has been paid to waterlogging compared with heat and drought stress. The authors did a great job in simulating the effect of waterlogging stress on barley production across global environments using the improved APSIM model. The method looks robust, and the findings are of importance to a wider community. Generally, the work is well organized, and the readability of the manuscript is also good.

Response: We appreciate your positive feedback on our manuscript.

Model performance.

1. The process basis for simulating waterlogging in the APSIM model shows significant advantages over many other crop models without the inclusion of soil waterlogging. It would be nice to discuss the model performance using available observed datasets. This is because yield reduction caused by waterlogging stress is generally from multiple aspects, e.g. delayed phenology (the improved APSIM shows this function), reduction in biomass, grain number etc.

Response: The advantages of and relative performance of APSIM compared with other waterlogging-enabled models simulating waterlogging stress and plant recovery is detailed at length in our previous peer-reviewed literature^{54,56}. Similarly, the performance of APSIM

in simulating the effects of waterlogging on phenology, biomass and yield components has been evaluated in our prior peer-reviewed work³⁹. Together, the results from these papers demonstrates that the variability within the simulated data was similar to that in observed data, indicating that the model conceptual design and parametrisation was adequate. For the sake of transparency – and because future readers may have similar questions – we have added text similar to that above to the methods.

Line 475-477

2. The evaluation of soil moisture dynamics and water table depth in the model, which is the basis for simulating waterlogging stress on crop growth. I can only see some simulated and observed results relevant to water table depth in the supplementary information. How about the model's performance on soil water dynamics?

Response: We document calibration in Supplementary Figure 12 and sensitivity analyses of modelled soil water dynamics on yield, phenology and leaf expansion in previous work⁵⁶. To minimise confounding effects associated with spatial and vertical variation in soil properties under field conditions, we calibrated the model using experimental data measured under controlled conditions³⁸. In concert with sensitivity analyses outcomes in the aforementioned paper, we believe that this is sufficient evidence of rigor in modelled soil water dynamics.

3. Soil waterlogging may occur for many reasons, including extreme rainfall events, intense rainfall in short periods, poor hydraulic conductivity, rising/perched water tables, or combinations of these factors. I also checked your previous publication (Liu et al., 2020 ERL) that calls for a better understanding of the model's sensitivity to hydraulic conductivity. Here you have examined 38 sites that have different soil properties in your study. How the sensitivity of those factors to waterlogging occurrence was addressed here?

Response: We agree; in fact, part of the reason we assessed 38 sites was so that our study encapsulated this natural variability in soil properties. We discuss avenues for waterlogging (and more) in the paper the reviewer mentions above. Regardless of the mechanism in which waterlogging occurs however, perceived impacts by plants is typically differential root-zone waterlogging that is bottom-up, top-down or lateral. We test our conceptual model design and integration within the APSIM framework by assessing the average number of days the root zone is saturated using sensitivity analyses⁵⁶ and factorial simulations in which the climate, management and environment type are varied³⁹. We show that sites with lower soil hydraulic conductivity, shallow watertable depths and higher growing season rainfall have higher frequencies of waterlogging, evidenced for example by the Arras sub-region in France (Fig. 2). The intermittent nature of waterlogging within and across seasons and sites was one of the key reasons we developed the approach for characterising common waterlogging stress patterns, allowing us to draw generalized insights through discrete outputs that integrate many thousand simulations.

Modelling waterlogging process

4. Your previous paper (Liu et al., 2020) suggests that the watertable depth seems never

exceeded the plant height. Heavy rainfall could cause inundation and submergence, which could cause crop failure. Does the APSIM model account for crop failure due to submergence?

Response: Indeed, heavy rainfall, excessive irrigation or lateral surface flows could cause submergence. We assume that the majority of waterlogging effects on plants occurs via oxygen deficit through root zone implications and as such, do not account for crop failure. This decision is well justified for two reasons. First, the vast majority of waterlogging across sites and seasons is highly transient, evidenced by data shown in Fig. 3. Second, our experimental measurements have shown that waterlogging intolerant barley genotypes were able to persist for waterlogging periods of up to two months without failing³⁸. We have added this justification to the methods.

Line 496-499

5. How are waterlogging-stress days defined in the improved model? and how the four waterlogging types were identified here? The information in Fig. 3 is not very clear to me.

Response: We added the following to the methods:

“Effects of waterlogging on photosynthesis and phenology (‘waterlogging-stress days’) were modelled using stress indices (oxdef_photo and oxdef_pheno) computed as a function of the fraction of roots waterlogged (oxdef_photo_rtfr). For oxdef_photo_rtfr levels of 0.8 and greater, oxdef_photo and oxdef_pheno linearly decreased; for oxdef_photo_rtfr levels less than 0.8, no stress was invoked, following experimental observations^{38, 78}.”

Line 541-546

We clarified the caption of Fig. 3 as follows:

Fig. 3 | Waterlogging (WL) stress patterns and frequencies and grain yields for the baseline (1985-2016), 2040 (2030-2059) and 2080 (2070-2099). Data shown for spring (a-c) and winter barley (d-f) across sites, sowing times and genotypes. Four key waterlogging stress patterns across sites and genotypes are depicted: stress patterns for spring barley include SW0 (minimal waterlogging); SW1 (low moderate-late waterlogging); SW2 (late-onset moderate waterlogging); SW3 (late-onset severe waterlogging) and winter barley WW0 (minimal waterlogging); WW1 (low early-onset waterlogging relieved later); WW2 (moderate early-onset waterlogging); WW3 (severe early-onset waterlogging). Boxplots indicate grain yields for spring and winter barley across sites and GCMs; box boundaries indicate the 25th and 75th percentiles across 27 GCMs, whiskers below and above the box indicate the 10th and 90th percentiles. Growth stages include the early juvenile phase (JV1, 10≤APSIM growth stage<21; late juvenile phase (JV2, 21≤APSIM growth stage<32); floral initiation to heading (FIN, 32≤APSIM growth stage<65); flowering to grain filling (FIN, 65≤APSIM growth stage<71; early grain filling (GF1, 71≤APSIM growth stage<80) and late grain filling (GF2, 80≤APSIM growth stage<87).

Line 223-231

6. Schematic of genotypic traits influenced by waterlogging in Fig. 1 is not intuitive. Please provide more descriptions of how those functions work in the model.

Response: In line with this comment, together with those from the handling Editor, we added the following to the methods:

“To account for genetic differences in waterlogging tolerance in APSIM, we invoke the function `y_oxdef_lim_photo`, where `y_oxdef_lim_photo` values of unity or nil equate to no stress or full stress, respectively. The third stage encompasses adaptation responses, the net result of which is a variable level of adaptation depending on waterlogging tolerance genetics. After the adaptation stage, genotypes tolerant to waterlogging tend to exhibit similar photosynthetic rates compared with before waterlogging, analogous to plants that grow aerenchyma after waterlogging³⁸, whereas genotypes sensitive to waterlogging can exhibit decreased growth after waterlogging events if `y_oxdef_lim_photo` remains less than unity. We did not mathematically transcribe a process for crop failure under waterlogging, because in the majority of cases, waterlogging is realised as a transient event (viz. Fig. 3) and our experimental work suggests that intolerant genotypes persist for up to two months of waterlogging without failing³⁸. These concepts were programmed into the source code of APSIM; the executable containing the modified source code and XML files are available online: <https://github.com/KeLiu7/Waterlogging-Barley>.”

Line 489-501

And:

“Effects of waterlogging on photosynthesis and phenology (‘waterlogging-stress days’) were modelled using stress indices (`oxdef_photo` and `oxdef_pheno`) computed as a function of the fraction of roots waterlogged (`oxdef_photo_rtfr`). For `oxdef_photo_rtfr` levels of 0.8 and greater, `oxdef_photo` and `oxdef_pheno` linearly decreased; for `oxdef_photo_rtfr` levels less than 0.8, no stress was invoked, following experimental observations^{38, 78}. Photosynthetic and phenological stress indices were also defined as a function of crop stage (`x_oxdef_stage_photo`, `x_oxdef_stage_pheno`), which is a significant advance on the majority of previous studies which assume that waterlogging stress depends only on the extent and duration of water-filled pore space⁷⁹. Part of the novelty of the current work is the delay in phenology associated with the duration of waterlogging and the crop stage/s in which it occurs. Waterlogging in early growth stages inhibits leaf appearance rate and tiller development and delays flowering. If waterlogging stress occurs during vegetative stages³⁸, plants may fully recover by grain-filling stages; if waterlogging occurs during flowering, plants cannot fully recover pre-waterlogging photosynthetic potential before maturity³⁸. Effects of waterlogging on phenology (`oxdef_pheno`) were similarly derived using information from environment-controlled experiments³⁸. The parameter `oxdef_pheno` was computed as a function of the fraction of roots waterlogged (`oxdef_pheno_rtfr`). For `oxdef_pheno_rtfr` levels of 0.8 or greater, `oxdef_pheno` linearly decreased to 0.8 until the soil is fully saturated; for `oxdef_pheno_rtfr` levels less than 0.8, no stress was invoked.

Line 541-558

On the other hand, the simulated results in the improved APSIM were much better compared with the default APSIM model. I'm wondering if the waterlogging functions in the improved model could emerge in other crop models, which will benefit the crop modelling community a lot. Please note this is not a real remark, but a point of potential discussion.

Response: Good suggestion; we are currently involved in the leadership team of an international agricultural waterlogging intercomparison project (AgMIP waterlogging <https://forms.gle/Q5eTQKbJJeQvDHmt5>) for which these improvements could be used in. The high level of detail re processes implicated in waterlogging together with the descriptions of the simple functional relationships we added to the methods (detailed in responses above) will aid other people in developing and building from our work in future.

7. Based on Fig. 2, it seems that winter barley has a greater yield penalty by waterlogging stress. I am sure that you are talking about sowing time here. Spring sowing has much less chance of waterlogging thus less difference between different models. As in Sup Table 2, spring barley refers to spring type barley and winter barley refers to winter type barley, the terms "spring sowing barley" and "autumn/winter sowing barley" should be used here.

Response: Thanks for this observation. To clarify, the caption of Fig. 2 was revised as follows:

"Impacts of waterlogging on yield under future climate (2040, 2080) relative to the historical baseline (1985-2016) for early and late sowing (ES, LS). a, c, Simulated yield differences under future climate with and without waterlogging (WL) for genotypes with early (spring sowing barley) or late maturity (autumn/winter sowing barley). b, d, simulated yields (pie charts; dark segments denote yield penalty) under late sowing for spring barley and early sowing for winter barley in 2040 (results for early or late sowing in 2040 and 2080 can be found in supplementary Fig. 2). Yields were simulated with APSIM using downscaled projections from 27 GCMs. Boxplots indicate simulated yield change across sites and GCMs; box boundaries indicate 25th and 75th percentiles, whiskers below and above each box denote the 10th and 90th percentiles, respectively. Green regions in the maps define predominant barley cropping areas."

Line 198-203

8. The typology in Fig. 3 is another main result of the paper. You seem to refer to each class of waterlogging as a "typology", which might be not very accurate. The abbreviation for WL is not explained in the caption.

Response: We agree the term 'typologies' could be confused and have replaced this with 'waterlogging stress patterns', which we believe is more intuitive.

The abbreviation 'WL' stands for "waterlogging" - we clarified this in the caption.

Indeed, Fig. 3 is a key result and part of the novelty of this work.

Line 223-231

9. Early sowing and late sowing should have consistent abbreviations. In Fig. 2, it was E and L, respectively and it was ES and LS in Fig. 4 (typos in the caption of Sup Fig 3). Also, please clarify the significant level, at 0.05 or 0.01.

Response: Amended as suggested throughout the manuscript.

The caption of Fig 4 was revised as follows:

“Fig. 4| Grain yield penalty and waterlogging stress patterns for the baseline (1985-2016), 2040 (2030-2059) and 2080 (2070-2099). Grain yield penalties are shown for spring (a) and winter (d) barley across sites and genotypes for relatively early or late sowing (ES, LS) at each site. Waterlogging stress patterns for spring barley include SW0 (minimal waterlogging); SW1 (low moderate-late waterlogging); SW2 (late-onset moderate waterlogging); SW3 (late-onset severe waterlogging) and winter barley, WW0 (minimal waterlogging); WW1 (low early-onset waterlogging relieved later); WW2 (moderate early-onset waterlogging); WW3 (severe early-onset waterlogging). Different letters in (b), (c), (e) and (f) indicate significant difference(s) in frequency of stress patterns between climate periods within waterlogging stress patterns (P<0.05). Error bars are standard errors of the mean. Boxplots indicate yield penalty for spring and winter barley across sites and GCMs; box boundaries indicate 25th and 75th percentiles, whiskers below and above the box indicate the 10th and 90th percentiles, respectively.”

Line 233-239

10. In Fig. 5, you said that altering sowing time coupled with the adoption of superior genetics resulted in further gains in yield. Why the results (yield benefit) are negative in some places? Also, is this correct that SEM was so small?

Response: The yield benefit of the waterlogging tolerant genotypes was strongly influenced by management and environment interactions. In Fig. 5, we simulated yields using 27 GCM, with each model generating different climate realisations. For some environments where waterlogging stress occurred in later growth phases, the benefit was minimal (i.e. both waterlogging tolerant genotypes and waterlogging-susceptible genotypes suffered high yield penalty) or slightly negative, as the reviewer rightly observes. Note however that such cases are in the minority.

As for the SEM, we regard the results as reasonable because the only difference between the two genotypes was their waterlogging tolerance (i.e. the values in this figure are SEM of a relative difference rather than SEM of the yield per se). We have clarified this in the caption of Fig. 5.

Discussion

11. Your results showed that the overall mean yield penalty from waterlogging increased from 3-11% (baseline conditions) to 6-14% (2040) and 10-20% (2080). The yield loss largely depends on the genotypes' sensitivity to waterlogging stress thus genotypic effects should be discussed (i.e. potentially under-estimated or over-estimated).

Response: Agree. We added the following text to the Discussion.

“Although we revealed multiple prospects for alleviating crop waterlogging under future climates, the variability in simulated yield and phenology responses under future climates highlights the importance of genotypic sensitivity to waterlogging stress. Across scenarios, mean yield penalty from waterlogging increased from 3-11% (baseline) to 6-14% (2040) and 10-20% (2080). Potential yield losses largely depend on genotypic sensitivity to waterlogging stress, in general with greater yield gains for tolerant genotypes of early sown (winter maturity) waterlogging tolerant genotypes, and the lowest gains for later sowing of (spring maturity) waterlogging tolerant genotypes (Fig. 5). We obtained genotypic parameters for waterlogging tolerance and phenology from previous empirical studies^{38,78} but additional parameters from local genotypes would help improve the rigor of projected changes under future climates. However, we emphasize that the relative difference between scenarios is more important than the absolute values in this study.”

Line 436-447

12. Authors should also discuss the limitation of this study. For example, only one model (APSIM) was used in this study. Martre et al. (2015) indicated that multi-model ensembles are much better than one. The limitations of multi-model ensembles and the possibility of transferring the waterlogging functions developed in APSIM to other models should be discussed.

Response: Agree. The following texts are added in the Discussion section.

“In this study, we used APSIM based on evidence that suggests that this farming systems framework is one of the most reliable models for simulating waterlogging dynamics^{54,56}. Increasingly, however, multi-model ensemble studies for predicting agroecological variables are becoming commonplace, associated with the rise of high-performance computing, big data and cloud analytics^{30, 31,57}. Some ensemble studies suggest that taking either the ensemble mean or median of simulated values provide more accurate estimates than any individual model when variables related to growth are considered^{58,59}. Indeed, the authors of the present study are working as part of an international research team in an Agricultural Modelling Intercomparison Project (AgMIP)⁵⁵ to test the applicability of our new approaches in a global study of crop waterlogging. This will allow us to scale our developments from barley to other genotypes, management options and environments using a range of models and, together with co-design as part of a community of AgMIP practitioners, improve the rigor of the approaches developed here”.

Line 402-413

13. At the end of the discussion, a paragraph explaining the uncertainty mentioned above of the research should be added.

Response: We added the following discussion:

“While we only used one crop model, we invoked projections from an ensemble of outputs from 27 global climate models (GCMs). This aspect could be construed as both a strength and a weakness; the former because the ensemble mean of climatic projections

should be more reliable than a projection from any one GCM (as discussed above), the latter because the variability in modelled outputs increases associated with greater variability in climatic realisations. Larger variability in outputs increases the uncertainty associated with the projection and can make results from such studies more difficult to comprehend in a rationally bounded way^{48,60}. In fact, such diversity in potential simulated results across sites, seasons, genotypes and management was a key reason we developed the new approach to cluster waterlogging stress patterns.

Of all GCM outputs, rainfall is perhaps the most uncertain. A key reason for using the data from 27 GCMs was to better quantify the spatio-temporal distribution of and variability in precipitation during the growing season. We downscaled the GCM datasets using the NASA/POWER gridded historical weather database⁶¹. However, previous work has shown that interpolated gridded data tends to be conducive to producing rainfall events that are smaller in quantum but more frequent, which can lead to lower surface runoff and higher soil evaporation⁶¹. In a crop model, this could reduce plant water and nitrogen uptake, resulting in propagation of errors that impact on variables such as biomass and yield. Using agricultural systems models with observed data before spatially interpolating point-based results may thus represent a more preferable approach for reducing uncertainty in model outputs. While the present study avoids the aforementioned issue associated with nitrogen uptake because nitrogen stress was not invoked, in practice, mineral nitrogen deficiencies associated with waterlogging may be present because waterlogging impacts on the ability of plant roots to uptake nutrients^{62,63}.”

Line 414-435

Materials and method

14. Authors collected some yield datasets of waterlogging experiments globally to evaluate the model performance. How do the authors collect the soil and climate datasets from each location? Please correct the title as “Data for waterlogging function parameterisation and validation”.

Response: *Daily data for maximum and minimum temperature, rainfall and solar radiation for 1985–2016 at each location were obtained from the National Aeronautics and Space Administration/Prediction of Worldwide Energy Resources (NASA/POWER)⁶⁷. NASA/POWER provides climate data at a horizontal resolution of 1° latitude-longitude. Yearly atmospheric CO₂ concentration [CO₂] for future periods were calculated using empirical equations that were obtained by nonlinear least-squares regression, based on the Shared Socio-economic Pathway 585 (SSP585), a business-as-usual (high) emission scenario (see Supplementary Methods). Soil parameters (soil texture, bulk density, pH, and organic carbon content etc.) were obtained from the International Soil Reference and Information Centre⁷¹.*

Line 588-596

Accessibility to both climate and soil data is shown in the ‘data availability’ section (<https://github.com/KeLiu7/Waterlogging-Barley>).

We changed the subtitle to “Experimental data used for parameterisation and evaluation” as we believe this is a more accurate reflection of the section content. We also changed all

mentions of “validation” to “evaluation” as models can only be evaluated and are not always valid.

Line 451

15. Authors proposed a good idea on crop response, adaptation and recovery from waterlogging for alternative genotypes. However, it is still confusing how the waterlogging-tolerant genotypes were simulated in the model. Are there any parameters to account for this? if yes, please provide more details in the “Impact of waterlogging tolerance genes on barley growth and development.”

Response: We added a significant amount of detail to the methods to clarify this point:

“To account for genetic differences in waterlogging tolerance in APSIM, we invoke the function `y_oxdef_lim_photo`, where `y_oxdef_lim_photo` values of unity or nil equate to no stress or full stress, respectively. The third stage encompasses adaptation responses, the net result of which is a variable level of adaptation depending on waterlogging tolerance genetics. After the adaptation stage, genotypes tolerant to waterlogging tend to exhibit similar photosynthetic rates compared with before waterlogging, analogous to plants that grow aerenchyma after waterlogging³⁸, whereas genotypes sensitive to waterlogging can exhibit decreased growth after waterlogging events if `y_oxdef_lim_photo` remains less than unity. We did not mathematically transcribe a process for crop failure under waterlogging, because in the majority of cases, waterlogging is realised as a transient event (viz. Fig. 3) and our experimental work suggests that intolerant genotypes persist for up to two months of waterlogging without failing³⁸. These concepts were programmed into the source code of APSIM; the executable containing the modified source code and XML files are available online: <https://github.com/KeLiu7/Waterlogging-Barley>.”

Line 489-501

“Waterlogging results in inhibition of processes in the mesophyll, photoassimilate transport in the phloem, gas conductance and thus reduces photosynthetic rate⁵³. In APSIM-Barley, these processes are modelled per unit ground area. Effects of waterlogging on photosynthesis and phenology (‘waterlogging-stress days’) were modelled using stress indices (`oxdef_photo` and `oxdef_pheno`) computed as a function of the fraction of roots waterlogged (`oxdef_photo_rtfr`). For `oxdef_photo_rtfr` levels of 0.8 and greater, `oxdef_photo` and `oxdef_pheno` linearly decreased; for `oxdef_photo_rtfr` levels less than 0.8, no stress was invoked, following experimental observations^{38,78}. Photosynthetic and phenological stress indices were also defined as a function of crop stage (`x_oxdef_stage_photo`, `x_oxdef_stage_pheno`), which is a significant advance on the majority of previous studies which assume that waterlogging stress depends only on the extent and duration of water-filled pore space⁷⁵. Part of the novelty of the current work is the delay in phenology associated with the duration of waterlogging and the crop stage/s in which it occurs. Waterlogging in early growth stages inhibits leaf appearance rate and tiller development and delays flowering. If waterlogging stress occurs during vegetative stages³⁸, plants may fully recover by grain-filling stages; if waterlogging occurs during flowering, plants cannot fully recover pre-waterlogging photosynthetic potential before

maturity³⁸. Effects of waterlogging on phenology (*oxdef_pheno*) were derived using information from environment-controlled experiments³⁸. The parameter *oxdef_pheno* was computed as a function of the fraction of roots waterlogged (*oxdef_pheno_rtfr*). For *oxdef_pheno_rtfr* levels of 0.8 or greater, *oxdef_pheno* linearly decreased to 0.8 until the soil is fully saturated; for *oxdef_pheno_rtfr* levels less than 0.8, no stress was invoked.

Line 539-558

To account for varying effects on phenology, we invoke the function *y_oxdef_lim_pheno* that is calculated according to crop stage (*x_oxdef_stage_pheno*, i.e. APSIM stage code). The *y_oxdef_lim_pheno* response function was adopted from our previous studies³⁸. For *y_oxdef_lim_pheno* levels less than 1, crop phenology is delayed, with *y_oxdef_lim_pheno* increasing from 0.65 at stage 4 to 0.95 to APSIM stage 5.5; for *y_oxdef_lim_pheno* levels greater than 1, grain-filling durations are truncated, with *y_oxdef_lim_pheno* increasing from 1.0 to 1.5 between APSIM stages 6 and 10. The delayed effect on phenology is only triggered before flowering (i.e. *x_oxdef_stage_pheno* between 1 and 6) and the grain filling duration reduction is triggered after flowering (i.e. *x_oxdef_stage_pheno* greater than or equal to 6). In general, the magnitude of delay is largely depended on the extent and duration of waterlogging stress, as well as its timing relative to crop development. The physiological basis for waterlogging-induced delays to phenology is discussed in our previous studies³⁸, with the rate of leaf emergence determining the duration between emergence and anthesis⁸⁰. Waterlogging in later growth stages causes premature flag leaf senescence and shortens the grain-filling period⁶³. Reduced grain growth in waterlogged plants is attributed to decreased post-anthesis carbon assimilation and culm reserves remobilised to grains^{81, 82}."

Line 560-575

16. For future climate scenarios, could you please explain why did you use monthly data instead of daily data at the first step (assuming that daily data are available for some of the mentioned GCM)?

Response: we added the following to the methods:

"There are several reasons we did not use daily data from GCMs in this study. First, not all GCMs provided daily climate data. Second, interpolation of daily GCM values can be error prone, especially for rainfall⁸¹. Third, weather variables such as radiation, precipitation, minimum and maximum temperature are often interdependent (e.g. rainy days are often cooler and have lower solar radiation); interpolation and bias-correction on a daily time-step can confound such interdependence⁸⁴. Instead, the approach we used (NWAI-WG bias-correction of monthly values) accounts for interdependency between climatic variables⁸⁵."

Line 609-615

17. This is a projection study. Please explain why the simulation climate periods of 2030-2059 and 2070-2099 were chosen.

Response: Our study was designed to investigate short-term and long-term implications of the climate crisis for waterlogging. We thus assumed two climate horizons (2040 and 2080)

and centred our simulations on each of these climatic windows (2030-2059 and 2070-2099), as the reviewer points out. Similar approaches have been used in other projection studies (*viz.* Wang et al. 2020), also justifying the methods we used here.

Wang, B., et al. Sources of uncertainty for wheat yield projections under future climate are site-specific. *Nat Food* 1, 720–728 (2020).

18. Please state what kind of R packages were used to do the K-means cluster in this study.

Response: Amended as follows:

K-means clustering was applied using the R package ‘stats’ (R Development Core Team, 2013).

Line 639-640

20. Add some explanation of the formula that how the atmospheric CO₂ concentrations were calculated.

Response: We added the following to the methods:

“Annual atmospheric concentrations [CO₂] were calculated for each year following the method⁸⁴ that used empirical equations obtained by nonlinear least-squares regression fitted to [CO₂] from the 27 GCMs (see Supplementary Methods).”

Line 594-596

21. I noticed that you compared the yield difference between the improved model and the default model. Did you run the model two times or set the waterlogging parameters at 0 to remove waterlogging effects? Please indicate if the default APSIM model includes waterlogging or not. This is important to evaluate the waterlogging effects on crop production.

Response: The default version of APSIM has waterlogging functions but not accounting for yield penalty under waterlogging stress.

To clarify this, we added the following to the methods:

“The default version of APSIM contains photosynthesis functions for waterlogging (oxdef_photo), but these do not account for the effects of waterlogging on phenological rate. To compare the improved version of APSIM (detailed below) with the default version, we ran the improved model twice: once with oxdef_photo set to 1, and again with these parameters using values detailed as below.”

Line 485-489

22. It’s okay to create hypothetical genotypes with factorial variations in traits to represent the local cultivars. However, not all modellers are familiar with APSIM model, thus more details on the basis for setting the parameter for spring and winter barley, respectively, would be helpful.

Response: Thanks for this suggestion. We added the following to the methods:

*“Genotypes were created such that lifecycles were in line with local sowing, flowering and maturity times⁷⁶. This was conducted by setting APSIM phenological parameters for vernalisation (*vern_sens*), photoperiod (*photop_sens*) and thermal time between emergence and the end of the juvenile phase (*tt_end_of_juvenile*). These parameters were chosen due to their high influence on crop flowering times⁴⁵. Winter barley requires greater exposure to cold temperature to evoke reproductive development, whereas spring barley flowers without a cold exposure precondition. In APSIM-Barley, *vern_sens* refers to vernalization representing cumulative cold temperature requirement to initiate reproductive development (range 0 to 5), while *photop_sens* refers to day length sensitivity (range 0-5); higher values denote greater sensitivity. We assigned *vern_sens* values based on maturity group (‘spring’ maturity = 1 and ‘winter’ maturity values of either 2.5 or 4, depending on vernalization requirement)⁷⁷. Similarly, *photop_sens* values were set to 1 for ‘spring’ maturity and ‘winter’ maturity values ranged from 2.5 to 4. *tt_end_of_juvenile* values were set to 400 for ‘spring’ maturity and ‘winter’ maturity values ranged from 400 to 750. We first parameterised *vern_sens* according to the maturity group, we then adjusted *photop_sens* and *tt_end_of_juvenile* until the simulated flowering days match with local flowering and maturity.”*

Line 522-537

Some other minor comments:

Page 4, Y or y in the function should be consistent.

Response: Amended all to ‘y’.

Line 145-149

Page 20, please provide the full name for AgMIP.

Response: Added:

Agricultural Modelling Intercomparison Project (AgMIP)

Line 409-410

Supplementary Table 1. The full name of GCM should be global circulation models.

Response: Amended as suggested

Supplementary Table 3, please use color blind friendly to make your Table accessible.

Response: We changed the colour scheme from red/green to red/blue.

Reviewer #3 (Remarks to the Author):

General comment

This paper demonstrated the potential of crop models for decision-making in an agronomy field un-explored such as crop waterlogging stress. It is original and it has great significance for the agronomy and food security fields.

Response: Thank you for your positive feedback on our work.

The work meets the expected standards in the crop modelling field except for one reason, which is crucial for model developers, it has a lack of code review by independent peers. Therefore, the main limitation of this work is that the presented results can have a bias related to the updated/modified Barley code which was not reviewed by independent researchers from the crop modelling community and property released (or at least it was not stated in the manuscript). I strongly suggest to update the code review status or wait until the updated Barley model is properly reviewed and released by the APSIM Initiative Panel before this paper can be published.

Response: In line with our responses to the previous reviewers above, we added significant detail to the methods to improve transparency of the conceptual design, parameterisation and evaluation conducted. We also provided complete access to all model inputs, outputs and underpinning algorithms in the code availability section. We did not develop these new algorithms with the intention of incorporation into future releases of APSIM. In fact, the purpose of the present paper was for peers to review the concepts and framework we used to formulate waterlogging responses. This is original research: it has not been published elsewhere. Before any algorithms can be incorporated into the APSIM core release, they first must demonstrate their value with the scientific community: this was part of the intent of the current study. As well, there are numerous other high-profile studies that have added new scripts to APSIM without going to the Reference Panel (e.g. Harrison et al. 2014; Zhao et al., 2022; Zhang et al., 2020; Archontoulis et al., 2016; Shi et al., 2022). Instead, scripts used in these papers underwent peer-review as part of the publication process. For the APSIM biochar model (Archontoulis et al. 2016), the script was first peer-reviewed and published before it was submitted to the reference panel and later incorporated into APSIM. This review process allows the global science community to critique development processes for APSIM and is the one of the main reasons APSIM is an internationally recognised platform for simulation of agricultural systems. Finally, it is worth noting that the purpose of embedding our algorithms within the APSIM source code was not so we could later integrate them into the default version of APSIM. Instead, we modified the source code so we could (1) test the validity of our new concepts within an integrated soil-plant-climate framework, (2) conceive a new approach for clustering waterlogging stress trajectories, and (3) simulate pathways for waterlogging adaptation on the global scale.

1. Harrison, M.T., Tardieu, F., Dong, Z., Messina, C.D., Hammer, G.L., 2014. "Characterizing drought stress and trait influence on maize yield under current and future conditions". *Global Change Biology* 20, 867-878.

1. Zhao et al. "Novel wheat varieties facilitate deep sowing to beat the heat of changing climates." *Nature Climate Change* 12.3 (2022): 291-296.

2. Zhang et al. "The contribution of spike photosynthesis to wheat yield needs to be considered in process-based crop models." *Field Crops Research* 257 (2020): 107931.
3. Archontoulis et al. "A model for mechanistic and system assessments of biochar effects on soils and crops and trade - offs." *GCB Bioenergy* 8.6 (2016): 1028-1045.
4. Shi et al. "Radiation use efficiency and biomass production of maize under optimal growth conditions in Northeast China." *Science of The Total Environment* 836 (2022): 155574.

Specific comments

L121. Is there an appropriated target stress pattern? Generally, the target is to avoid the stress to increase yield production. It is difficult to think that a stress pattern can be targeted.

Response: Stress patterns depend on season, environment, genotype and management. When simulated over the long term and clustered into common groups, stress frequencies at each manifest for each site. For example, we found that late season waterlogging stress was and will be more common in France compared with other regions (Supplementary Information Fig. 3). Knowledge of long-term frequencies of such stress patterns at each site can be used to determine whether or not there is a need to adapt, for example selection of waterlogging tolerant genotypes or changes in sowing time.

To clarify this, we changed the text to:

"Armed with the knowledge of stress prevalence and pattern using this method, decision-makers can more intuitively identify the most appropriate adaptation within stress patterns that are more probable for their environment, but can also transfer adaptations across regions within any given stress type^{34,37}".

Line 120-123

L442. Although the magnitude of the simulation analysis was done by the author is massive, a proper review of the updated model code is required before publication. If the updated Barley APSIM model was already reviewed by peers and released in the APSIM GitHub repository, please share the link to the code in this manuscript. Otherwise, the authors should wait to get the code reviewed by an independent group of researchers/programmers before publication of results using the updated model. As I understood an APSIM crop model (such as the updated Barley model presented here) is a prototype if it was not reviewed and approved by the APSIM Initiative Panel.

Response: Please see our first response to this reviewer.

L449. Why did you use only soil and not climate to prioritized sites?

Response: We considered both soil types and climates to prioritise sites; within each site, predominant soil types were selected. We have clarified the methods in this regard.

Line 503-516

L478. The authors name APSIM version 7.9 to explain the phenology module in the Barley model. However, they did not explicitly name the model version used for this study which makes difficult to check code availability in GitHub. Although a GitHub repo was shared (<https://github.com/KeLiu7/Waterlogging-Barley>) it does not show the updated APSIM Barley model. The code shared is not providing any information about the updates implemented in the model.

Response: In line with *Nature* guidelines, we uploaded the code, explanations and instructions for running the improved model onto GitHub:
<https://github.com/KeLiu7/Waterlogging-Barley>.

L483. Which method did you use for genotype parametrization in the model? The authors stated they ‘match’ the thermal time between emergence and maturity for each site by adjusting phenological parameters. However, the model bias (observed - predicted) can be generated for any of these parameters. How you know the contribution of each parameter to the total model bias?

Response: For each site, we created virtual genotypes by altering three sensitive parameters influencing phenology in APSIM (based on previous peer-reviewed work) such that crop lifecycles were in line with locally observed data. As suggested by the previous reviewer, this approach is acceptable and has been used in other global studies.

We added the following to the methods:

*“Genotypes were created such that lifecycles were in line with local sowing, flowering and maturity times⁷⁶. This was conducted by setting APSIM phenological parameters for vernalisation (*vern_sens*), photoperiod (*photop_sens*) and thermal time between emergence and the end of the juvenile phase (*tt_end_of_juvenile*). These parameters were chosen due to their high influence on crop flowering times⁴⁵. Winter barley requires greater exposure to cold temperature to evoke reproductive development, whereas spring barley flowers without a cold exposure precondition. In APSIM-Barley, *vern_sens* refers to vernalization representing cumulative cold temperature requirement to initiate reproductive development (range 0 to 5), while *photop_sens* refers to day length sensitivity (range 0-5); higher values denote greater sensitivity. We assigned *vern_sens* values based on maturity group (‘spring’ maturity = 1 and ‘winter’ maturity values of either 2.5 or 4, depending on vernalization requirement)⁷⁷. Similarly, *photop_sens* values were set to 1 for ‘spring’ maturity and ‘winter’ maturity values ranged from 2.5 to 4. *tt_end_of_juvenile* values were set to 400 for ‘spring’ maturity and ‘winter’ maturity values ranged from 400 to 750. We first parameterised *vern_sens* according to the maturity group, we then adjusted *photop_sens* and *tt_end_of_juvenile* until the simulated flowering days match with local flowering and maturity days.”*

Line 522-537

L524. Accordingly with the authors a reset function was not applied every year, therefore there is sequence/rotational effect, i.e., the water/N and C scenario at the end of a crop affects the water/N and C dynamics for the following crop. How did you count for the effect of rotation in your model simulations?

Response: Simulations were in fact reset each year to avoid potentially confounding or cascading effects associated with continuous simulations.

We clarified the methods as follows:

Simulations were run from 1985 to 2100; soil conditions were reset annually at sowing to prevent potential 'carry-over' effects from previous seasons.

Line 617-618

L525. In some regions of the world (e.g., NSW Australia), sowing date is defined by soil moisture content. During dry years, it is expected to have less sown area or in some cases crop failure due to lack of water availability for emergence. APSIM can model this behaviour. Why did you not apply variable sowing date based on water availability? This will produce more realistic long-term yield predictions based on interannual rainfall variability under future climates.

Response: One of the waterlogging adaptations we examined was sowing time (relatively early or late compared with long term sowing times for each site). Using a variable sowing time rule would confound diagnosis of the effect caused by this treatment and obscure identification of common waterlogging stress patterns across sites. To ensure consistency of emergence times across sites, initial plant available water was reset to 15 mm at sowing. We have clarified these details in the methods.

Reviewer #4 (Remarks to the Author):

The current study quantifies the effects of waterlogging on barley at different sites across the globe by developing a new modeling routine. In addition, they tested the potential of waterlogging tolerant cultivars with early and late maturity characteristics combined with shifting the sowing dates of spring and winter barley as adaptation strategies for climate change using the adjusted waterlogging module. The authors suggested that the developed pipeline can also be applied to other crops and environments. They projected yield reduction due to waterlogging would be between 10% and 20% by 2080. However, the CO₂ fertilization effects can overcompensate for the adverse impacts of climate change with or without considering waterlogging response in the model for most of the study locations. Shifting the sowing date together using resilient cultivars would significantly decline the negative effects of waterlogging. There are relevant research questions as well as an essential topic addressed in this manuscript. Recent studies such as Webber et al., 2020 indicated the importance of more robust consideration of waterlogging in process-based

crop models employed for impact assessment studies. Therefore, the significance of the research is undoubtedly, but the methodology is not novel. Excellent writing is evident in the manuscript. Aside from that, the presenting items are clear and informative.

Response: Thank you for your positive feedback.

Some issues regarding the model development, mechanism understanding, and assumptions for future projections need to be addressed before publishing:

- Model development: It is unclear (or maybe I did not get it!) how and based on which physiological mechanisms the waterlogging effects on phenology were implemented in the model. (a) Is the timing of the specific phenological stages advanced or delayed under waterlogging (written in the methods section)? If yes, what are the physiological bases for it? The authors referred to their published studies (Liu et al., 2020 (discussed in section 4.3); Liu et al., 2021) as a base for the new modeling routine. However, I did not find a concrete physiological base for phenology response in that study that can implement in a process-based model. Climate change can advance the timing of sensitive periods such as flowering, which can alter the overlap between waterlogging period and those phenological stages, but it has nothing to do with the direct response of phenology to waterlogging.

Response: We added the following to the methods:

“Waterlogging in early growth stages inhibits leaf appearance rate and tiller development and delays flowering. If waterlogging stress occurs during vegetative stages³⁸, plants may fully recover by grain-filling stages; if waterlogging occurs during flowering, plants cannot fully recover pre-waterlogging photosynthetic potential before maturity³⁸. Effects of waterlogging on phenology (oxdef_pheno) were derived using information from environment-controlled experiments³⁸. The parameter oxdef_pheno was computed as a function of the fraction of roots waterlogged (oxdef_pheno_rtfr). For oxdef_pheno_rtfr levels of 0.8 or greater, oxdef_pheno linearly decreased to 0.8 until the soil is fully saturated; for oxdef_pheno_rtfr levels less than 0.8, no stress was invoked.

To account for varying effects on phenology, we invoke the function y_oxdef_lim_pheno that is calculated according to crop stage (x_oxdef_stage_pheno, i.e. APSIM stage code). The y_oxdef_lim_pheno response function was adopted from our previous studies³⁸. For y_oxdef_lim_pheno levels less than 1, crop phenology is delayed, with y_oxdef_lim_pheno increasing from 0.65 at stage 4 to 0.95 to APSIM stage 5.5; for y_oxdef_lim_pheno levels greater than 1, grain-filling durations are truncated, with y_oxdef_lim_pheno increasing from 1.0 to 1.5 between APSIM stages 6 and 10. The delayed effect on phenology is only triggered before flowering (i.e. x_oxdef_stage_pheno between 1 and 6) and the grain filling duration reduction is triggered after flowering (i.e. x_oxdef_stage_pheno greater than or equal to 6). In general, the magnitude of delay is largely depended on the extent and duration of waterlogging stress, as well as its timing relative to crop development. The physiological basis for waterlogging-induced delays to phenology is discussed in our previous

studies³⁸, with the rate of leaf emergence determining the duration between emergence and anthesis⁸⁰. Waterlogging in later growth stages causes premature flag leaf senescence and shortens the grain-filling period⁶³. Reduced grain growth in waterlogged plants is attributed to decreased post-anthesis carbon assimilation and culm reserves remobilised to grains^{81, 82}.”

Line 550-575

(b) Whether different sensitivity to waterlogging is implemented depending on the phenological stage (that is, the sense getting from the main text)? This case is not new and was available on the old codes of APSIM. As far as I remember, there was an aeration deficit factor depending on the phenological stage in APSIM, which can linearly reduce the plant growth rate under waterlogging (high sensitivity in early growth stages and lower on maturity). Please check Asseng et al., 1997-figure 1.

Response: We modelled waterlogging stress as a function of both water-filled pore space and stage based experimental observations published in our past peer-reviewed literature^{54,56}.

The default version of APSIM does account for differential stage-dependent effects of waterlogging on growth as outlined in the Fig. 1 of Asseng et al (1997), but the sensitivity to waterlogging stress is not reasonable there. Please see the two Figures below that confirmed this (left figure is from our recent review³⁸ and right figure is from Webber et al., 2022, Agric Ecosyst Environ).

Figure 1: Crop sensitivity (AFS) to aeration deficit.

The novelty of our work is (1) updated sensitivity to waterlogging according to our previous experiment, (2) the *delay in phenology* associated with waterlogging, such that waterlogging impacts on not just photosynthesis, but also *phenology*. We have outlined other differences between the default version and improved version of APSIM in response to the second reviewer and have clarified this in the methods.

Figure 3. Effects of the timing of waterlogging relative to phenology on crop yield reduction. Example is shown for cereals. Color chart indicates the degree of yield loss. Abbreviations: AP, awn primordium; Al, anthesis; BGF, begin grain filling; CI, collar initiation; DR, double ridge; Em, seedling emergence; GS, growth stage; Hd, heading time; Hv, harvest; PM, physiological maturity; Sw, sowing.

Left Figure: Liu, K., Harrison, M. T., Shabala, S., Meinke, H., Ahmed, I., Zhang, Y., ... & Zhou, M. (2020). The state of the art in modeling waterlogging impacts on plants: what do we know and what do we need to know. *Earth's Future*, 8(12), e2020EF001801.

Right Figure :Webber, H., Rezaei, E. E., Ryo, M., & Ewert, F. (2022). Framework to guide modeling single and multiple abiotic stresses in arable crops. *Agriculture, Ecosystems & Environment*, 340, 108179.

Please indicate how exactly phenology considers in the new modeling routine, including the physiological mechanism behind the crop response, to make it clearer to readers from the current manuscript without the need to read at least two other manuscripts to understand the methodology.

Response: To clarify, the following was added to the methods:

“Waterlogging in early growth stages inhibits leaf appearance rate and tiller development and delays flowering. If waterlogging stress occurs during vegetative stages³⁸, plants may fully recover by grain-filling stages; if waterlogging occurs during flowering, plants cannot fully recover pre-waterlogging photosynthetic potential before maturity³⁸. Effects of waterlogging on phenology (oxdef_pheno) were derived using information from environment-controlled experiments³⁸. The parameter oxdef_pheno was computed as a function of the fraction of roots waterlogged (oxdef_pheno_rtfr). For oxdef_pheno_rtfr levels of 0.8 or greater, oxdef_pheno linearly decreased to 0.8 until the soil is fully saturated; for oxdef_pheno_rtfr levels less than 0.8, no stress was invoked.

To account for varying effects on phenology, we invoke the function y_oxdef_lim_pheno that is calculated according to crop stage (x_oxdef_stage_pheno, i.e. APSIM stage code). The y_oxdef_lim_pheno response function was adopted from our previous studies³⁸. For y_oxdef_lim_pheno levels less than 1, crop phenology is delayed, with y_oxdef_lim_pheno increasing from 0.65 at stage 4 to 0.95 to APSIM stage 5.5; for y_oxdef_lim_pheno levels greater than 1, grain-filling durations are truncated, with y_oxdef_lim_pheno increasing from 1.0 to 1.5 between APSIM stages 6 and 10. The delayed effect on phenology is only triggered before flowering (i.e. x_oxdef_stage_pheno between 1 and 6) and the grain filling duration reduction is triggered after flowering (i.e. x_oxdef_stage_pheno greater than or equal to 6). In general, the magnitude of delay is largely depended on the extent and

duration of waterlogging stress, as well as its timing relative to crop development. The physiological basis for waterlogging-induced delays to phenology is discussed in our previous studies³⁸, with the rate of leaf emergence determining the duration between emergence and anthesis⁸⁰. Waterlogging in later growth stages causes premature flag leaf senescence and shortens the grain-filling period⁶³. Reduced grain growth in waterlogged plants is attributed to decreased post-anthesis carbon assimilation and culm reserves remobilised to grains^{81, 82}.”

Line 550-575

What about photosynthesis's response to waterlogging? In figure 1, the authors mentioned radiation use efficiency (RUE) and photosynthesis in parenthesis. Those of two different modeling terminologies for converting intercepted radiation to biomass. Please be very specific about what exactly influences by waterlogging in the new routine. It also surprised me why the authors did not mention the effects of waterlogging on transpiration (due to stomatal closure) as the most commonly known crop response to waterlogging in other models. Do you have such a response in your APSIM version? Modeling of waterlogging is not well developed, such as drought modeling (it has some reasons I explain below), but we have well-tested robust routines in other models, such as DRAINMOD (Skaggs et al., 2012) and SWAGMAN Destiny (Yang et al., 2016) which consider not only photosynthesis response but also transpiration and leaf area expansion. Why do we need to develop a new routine with fewer processes to consider?

Response: We have clarified the terminology regarding RUE and photosynthesis throughout the paper. We added the following text to the methods:

“Waterlogging results in inhibition of processes in the mesophyll, photoassimilate transport in the phloem, gas conductance and thus reduces photosynthetic rate⁵³. In APSIM-Barley, these processes are modelled per unit ground area. Effects of waterlogging on photosynthesis and phenology (‘waterlogging-stress days’) were modelled using stress indices (oxdef_photo and oxdef_pheno) computed as a function of the fraction of roots waterlogged (oxdef_photo_rtfr). For oxdef_photo_rtfr levels of 0.8 and greater, oxdef_photo and oxdef_pheno linearly decreased; for oxdef_photo_rtfr levels less than 0.8, no stress was invoked, following experimental observations^{38,78}. Photosynthetic and phenological stress indices were defined as a function of crop stage (x_oxdef_stage_photo, x_oxdef_stage_pheno), which is a significant advance on the majority of previous studies which assume that waterlogging stress depends only on the extent and duration of water-filled pore space⁷⁹. Part of the novelty of this work is the delay in phenology associated with the duration of waterlogging and the crop stage/s in which it occurs. Waterlogging in early growth stages inhibits leaf appearance rate and tiller development and delays flowering. If waterlogging stress occurs during vegetative stages³⁸, plants may fully recover by grain-filling stages; if waterlogging occurs during flowering, plants cannot fully recover pre-waterlogging photosynthetic potential before maturity³⁸. Effects of waterlogging on phenology (oxdef_pheno) were derived using information from environment-controlled experiments³⁸. The parameter oxdef_pheno was computed as a function of the fraction of roots waterlogged (oxdef_pheno_rtfr). For oxdef_pheno_rtfr levels of 0.8 or greater,

oxdef_pheno linearly decreased to 0.8 until the soil is fully saturated; for oxdef_pheno_rtrf levels less than 0.8, no stress was invoked."

Line 539-558

In modelling crop responses to waterlogging, multiple biological and physical processes interact dynamically. Ultimately, we are interested in the development of simple but robust process-based approaches that reproduce observed phenomena. Recent reviews have found that APSIM is one of the most appropriate models for simulating crop waterlogging^{54, 56}. We built on this work, incorporating stage-dependent phenological implications caused by waterlogging. Our motivation was underpinned by parsimony: we sought to alter as few processes as possible relative to the default version of APSIM. Because the model is dynamic, effects of waterlogging on photosynthesis later flow through to reduced growth and thus, reduced transpiration and leaf area accumulation. Effects of waterlogging on transpiration and leaf area accumulation are thus emergent properties. A fruitful avenue for future work may be a comparison of transpiration and leaf area development from APSIM with that of other models (e.g. SWAGMAN and DRAINMOD).

The modeling of waterlogging is less developed because of the complexity of driving factors. To accurately simulate the impacts of waterlogging, local scale heterogeneity in topography, soils, severe compaction below the plough layer, and functional drainage must be taken into account, which can influence soil workability, crop establishment, and even nutrient leaching due to ponding versus runoff. How did APSIM consider those factors in a global analysis?

Response: We agree, waterlogging is an extremely wicked problem. This study does not propose all of the answers, but instead provides a step forward in scientific understanding of comparisons of waterlogging across regions, adaptations that may be plausibly employed (and those that should be jettisoned), as well as a new method for clustering water stress trajectories. This method could be used for any crop type or stress to integrate manifold outputs from crop models, distilling big data into discrete and intuitive categories. As well, it was the *relative differences* between climate scenarios and adaptations (genotype, sowing time, waterlogging tolerance) at the global scale was the key focus of this study, rather than the *absolute values* of the simulated outputs per se.

- Model calibration and validation: The parametrization and testing of the model are explained in a relatively general way, making it challenging to review the reliability of those processes. The authors mentioned in the text they only have phenology data for one experiment (line 551) and used only yield for parametrization of other experiments. This would substantially increase the risk of getting the right results for the wrong reasons. The model development and phenology response are the core of current research therefore, the parametrization for phenology should be carefully treated.

Response: We have elaborated at length on parameterisation and validation in response to previous reviewer questions, and have included more detail in the methods on this aspect.

Please see responses above.

- Limitations: The limitation of the current study needs to be clearly discussed. I suggest classifying them in two directions as input uncertainties and limitations in crop processes. The future climate projections are extremely uncertain regarding the temporal distribution of precipitation during the growing season, which is fundamental to assessing waterlogging risk therefore, we need to be careful in concluding waterlogging intensity for future windows such as the 2080s. The crop processes, such as early acclimation to waterlogging (as would be the case for winter barley in figure 3 d-f) (Herzog et al., 2015) or an increase in assimilate remobilization (Li et al., 2013) due to waterlogging not implemented in the model, would significantly change the results. The nitrogen stress was also switched off in the model execution however, it is against the nature of waterlogging since the roots under stress lost their nutrient uptake functionality. Please mention such limitations in discussing the results.

Response: We added discussion of the limitations the reviewer mentions (as well as those requested by previous reviewers). We included the following in the discussion:

"In this study we used APSIM based on evidence that suggests this farming systems framework is one of the most reliable in simulating waterlogging dynamics^{54, 56}. Increasingly, however, multi-model ensemble studies for predicting agroecological variables are becoming commonplace, associated with the rise of high-performance computing, big data and cloud analytics^{30,31,57}. Some ensemble studies suggest that taking either the ensemble mean or median of simulated values provide more accurate estimates than any individual model when variables related to growth are considered^{58,59}. Indeed, the authors of the present study are working as part of an international research team in an Agricultural Modelling Intercomparison Project (AgMIP)⁵⁵ to test the applicability of our new approaches in a global study of crop waterlogging. This will allow us to scale our developments from barley to other genotypes, management options and environments using a range of models and, together with co-design as part of a community of AgMIP practitioners, improve the rigor of the approaches developed here.

While we only used one crop model, we invoked projections from an ensemble of outputs from 27 global climate models (GCMs). This aspect could be construed as both a strength and a weakness; the former because the ensemble mean of climatic projections should be more reliable than a projection from any one GCM (as discussed above), the latter because the variability in modelled outputs increases associated with greater variability in climatic realisations. Larger variability in outputs increases the uncertainty associated with the projection and can make results from such studies more difficult to comprehend in a rationally bounded way^{49, 60}. In fact, such diversity in potential simulated results across sites, seasons, genotypes and management was a key reason we developed the new approach to cluster waterlogging stress patterns.

Of all GCM outputs, rainfall is perhaps the most uncertain. A key reason for using the data from 27 GCMs was to better quantify the spatio-temporal distribution of and variability in precipitation during the growing season. We downscaled the GCM datasets using the NASA/POWER gridded historical weather database⁶¹. However, previous work has shown that interpolated gridded data tends to be conducive to producing rainfall events that are smaller

in quantum but more frequent, which can lead to lower surface runoff and higher soil evaporation⁶¹. In a crop model, this could reduce plant water and nitrogen uptake, resulting in propagation of errors that impact on variables such as biomass and yield. Using agricultural systems models with observed data before spatially interpolating point-based results may thus represent a more preferable approach for reducing uncertainty in model outputs. While the present study avoids the aforementioned issue associated with nitrogen uptake because nitrogen stress was not invoked, in practice, mineral nitrogen deficiencies associated with waterlogging may be present because waterlogging impacts on the ability of plant roots to uptake nutrients^{62,63}.

Although we revealed multiple prospects for alleviating crop waterlogging under future climates, the variability in simulated yield and phenology responses under future climates highlights the importance of genotypic sensitivity to waterlogging stress. Across scenarios, mean yield penalty from waterlogging increased from 3-11% (baseline) to 6-14% (2040) and 10-20% (2080). Potential yield losses largely depend on genotypic sensitivity to waterlogging stress, in general with greater yield gains for tolerant genotypes of early sown (winter maturity) waterlogging tolerant genotypes, and the lowest gains for later sowing of (spring maturity) waterlogging tolerant genotypes (Fig. 5). We obtained genotypic parameters for waterlogging tolerance and phenology from previous empirical studies^{38,78} but additional parameters from local genotypes would help improve the rigor of projected changes under future climates. However, we emphasize that the relative difference between scenarios is more important than the absolute values in this study.”

Line 402-447

Minor issues:

- Please carefully define the difference between waterlogging and flooding in the text.

Response: Amended as follows:

Soil waterlogging occurs when soils are saturated and plant roots cannot respire, while flooding refers to excessive surface water accumulation⁶². Waterlogging may be present without surface flooding. Waterlogging can be caused by extreme rainfall events, prolonged seasonal rainfall, poor soil hydraulic conductivity, lateral surface and/or groundwater flows, rising/perched water tables, improper irrigation or combinations of these factors⁵³. Despite the diversity of ways in which waterlogging can occur, the ultimate result is oxygen levels in pore spaces that are insufficient for plant roots to adequately respire⁶⁹.

Line 503-509

Line 547-551

- Did you only consider rainfed systems or you had irrigated barley as well?

Response: Here we considered only rainfed systems as they are more widespread.

References

- Webber et al., 2020. No perfect storm for crop yield failure in Germany. Environmental Research Letters, 15: 104012.

- Asseng et al., 1997. Simulation of perched water-tables in a duplex soil. Proceedings of MODSIM '97, International Congress on Modelling and Simulation, Hobart, Tasmania, Australia, 8–11 December 1997.
- Skaggs et al., 2012. Drainmod: model use, calibration, and validation. Transactions of the ASABE.
- Yang et al., 2016. Prediction of salt transport in different soil textures under drip irrigation in an arid zone using the SWAGMAN Destiny model. *Soil Research*, 54(7): 869-879.
- Liu et al., 2020. Genetic factors increasing barley grain yields under soil waterlogging. *Food and Energy Security* 2020, 9(4): e238.
- Liu et al., 2021. Climate change shifts forward flowering and reduces crop waterlogging stress. *Environmental Research Letters*, 16(9): 094017.
- Herzog et al., 2015. Mechanisms of waterlogging tolerance in wheat – a review of root and shoot physiology. *Plant, Cell & Environment*, 39: 1068-1086.
- Li et al., 2013. Carbohydrates Accumulation and Remobilization in Wheat Plants as Influenced by Combined Waterlogging and Shading Stress During Grain Filling. *Journal of Agronomy and Crop Science*, 199: 38-48.

Reviewers' Comments:

Reviewer #1:

Remarks to the Author:

I am pleased with the implementation of my previous recommendations. I think the manuscript is now reproducible in its methodology and also the overall edits have helped to understand the methodology and results much better.

Reviewer #2:

Remarks to the Author:

I'm basically satisfied with the revisions prepared by the authors, so I suggest to accept it.

Reviewer #3:

Remarks to the Author:

The authors carefully replied to all co-authors and provided justifications to support each of the reviewer's comments. The updated manuscript was considerably improved and provides now a more clear methodology. Therefore, I accept the manuscript in the current form.

Reviewer #4:

Remarks to the Author:

The authors clearly and satisfactorily responded to my comments, and I have no further remarks. I enjoyed reading their manuscript and am sure that it would greatly contribute to our current understanding and better capture waterlogging effects on crop yield for impact assessment studies.

REVIEWERS' COMMENTS

Reviewer #1 (Remarks to the Author):

I am pleased with the implementation of my previous recommendations. I think the manuscript is now reproducible in its methodology and also the overall edits have helped to understand the methodology and results much better.

Response: We appreciate your positive feedback and thank you again for reviewing our manuscript.

Reviewer #2 (Remarks to the Author):

I'm basically satisfied with the revisions prepared by the authors, so I suggest to accept it.

Response: Many thanks for your constructive comments and thank you again for reviewing our manuscript.

Reviewer #3 (Remarks to the Author):

The authors carefully replied to all co-authors and provided justifications to support each of the reviewer's comments. The updated manuscript was considerably improved and provides now a more clear methodology. Therefore, I accept the manuscript in the current form.

Response: Thank you for your thoughts and the constructive suggestions. These have helped improve the clarity and rigor of our manuscript.

Reviewer #4 (Remarks to the Author):

The authors clearly and satisfactorily responded to my comments, and I have no further remarks. I enjoyed reading their manuscript and am sure that it would greatly contribute to our current understanding and better capture waterlogging effects on crop yield for impact assessment studies.

Response: We appreciate your positive feedback and thank you again for helping us improve our manuscript.